# Versican accumulation drives Nos2 induction and aortic disease in Marfan syndrome via Akt activation

María Jesús Ruiz-Rodríguez [ID][1,2], Jorge Oller[2,3,9], Sara Martínez-Martínez[1,2,9], Iván Alarcón-Ruiz [ID][1,2], Marta Toral [ID][1,2], Yilin Sun [ID][4], Ángel Colmenar[1,2], María José Méndez-Olivares [ID][1,2], Dolores López-Maderuelo[1,2], Christine B Kern [ID][5], J Francisco Nistal [ID][2,6], Arturo Evangelista[7], Gisela Teixido-Tura [ID][2,8], Miguel R Campanero [ID][2,4,10✉] & Juan Miguel Redondo [ID][1,2,4,10✉]

## Abstract

**Thoracic aortic aneurysm and dissection (TAAD) is a life-threatening condition associated with Marfan syndrome (MFS), a disease caused by fibrillin-1 gene mutations. While various conditions causing TAAD exhibit aortic accumulation of the proteoglycans versican (Vcan) and aggrecan (Acan), it is unclear whether these ECM proteins are involved in aortic disease. Here, we find that Vcan, but not Acan, accumulated in $Fbn1^{C1041G/+}$ aortas, a mouse model of MFS. $Vcan$ haploinsufficiency protected MFS mice against aortic dilation, and its silencing reverted aortic disease by reducing Nos2 protein expression. Our results suggest that Acan is not an essential contributor to MFS aortopathy. We further demonstrate that Vcan triggers Akt activation and that pharmacological Akt pathway inhibition rapidly regresses aortic dilation and Nos2 expression in MFS mice. Analysis of aortic tissue from MFS human patients revealed accumulation of VCAN and elevated pAKT-S473 staining. Together, these findings reveal that Vcan plays a causative role in MFS aortic disease in vivo by inducing Nos2 via Akt activation and identify Akt signaling pathway components as candidate therapeutic targets.**

**Keywords** Akt; Aortic Aneurysm; Marfan Syndrome; Nos2; Versican
**Subject Categories** Molecular Biology of Disease; Vascular Biology & Angiogenesis

## Introduction

Thoracic aortic aneurysms (TAA) progressively enlarge, increasing the risk of aortic dissection (AD), which is often lethal. TAA and dissections (TAAD) are associated both with systemic connective tissue disorders such as Marfan syndrome (MFS) and with non-syndromic disease (Ostberg et al, 2020). MFS is an inherited autosomal dominant disease caused by mutations in the fibrillin-1 gene (*FBN1*). Fibrillin-1 protein is an important microfibrillar component of the extracellular matrix (ECM) (Sakai et al, 2016) that contributes to the formation and maturation of elastic fibers (Ramirez and Dietz, 2007) and regulates TGF-β signaling by modulating TGF-β bioavailability within the ECM (Ramirez et al, 2018). *FBN1* mutations lead to elastic lamellae fragmentation and the deposition of collagen and proteoglycan (PG) in the ECM (Romaniello et al, 2014). These alterations are hallmarks of medial degeneration that compromise the structural integrity of the vascular wall, predisposing the aorta to AD. Life expectancy of untreated MFS patients is limited by aortic catastrophic events, which are the main cause of morbidity and mortality in these patients. Current management of MFS involves close follow-up, pharmacological treatment to slow aortic growth, and prophylactic surgery to prevent AD and premature death (Bossone and Eagle, 2021). Unfortunately, medical treatments do not efficiently slow aortic growth, leaving prophylactic surgery as the only effective therapy (Bossone and Eagle, 2021).

PGs, a major component of the ECM, are proteins covalently linked to glycosaminoglycans (GAGs). Versican (VCAN) and aggrecan (ACAN) are large extracellular PGs highly expressed in the aorta (Stanton et al, 2011; Suna et al, 2018). Under physiological conditions, PGs confer compression resistance to the aorta and regulate residual stress in the aortic wall, thereby helping to maintain proper mechanical and biological functions of the medial layer (Azeloglu et al, 2008). A pathological accumulation of PGs in the aortic medial layer increases intralamellar swelling pressure, locally raising aortic stress and disrupting connections between VSMCs and the elastic lamina, which results in local delamination and an aortic wall vulnerable to dissection (Humphrey 2013; Roccabianca et al, 2014a; Roccabianca et al, 2014b). In this context,

[1]Gene Regulation in Cardiovascular Remodeling and Inflammation Group, Centro Nacional de Investigaciones Cardiovasculares (CNIC), Madrid 28029, Spain. [2]Centro de Investigación Biomédica en Red de Enfermedades Cardiovasculares (CIBERCV), Madrid, Spain. [3]Laboratory of Vascular Pathology, Hospital IIS-Fundación Jiménez Díaz, 28040 Madrid, Spain. [4]Cell-Cell Communication & Inflammation Unit, Centro de Biología Molecular Severo Ochoa (CBMSO), Consejo Superior de Investigaciones Científicas-Universidad Autónoma de Madrid, Madrid 28049, Spain. [5]Medical University of South Carolina (MUSC), Charleston, SC 29425, USA. [6]Cardiovascular Surgery, Hospital Universitario Marqués de Valdecilla, Instituto de Investigación Valdecilla (IDIVAL), Facultad de Medicina, Universidad de Cantabria, Santander 39005, Spain. [7]Teknon Medical Centre-Quironsalud. Heart Institute, Barcelona, Spain. [8]Department of Cardiology, Hospital Universitari Vall d'Hebron (VHIR), Barcelona 08035, Spain. [9]These authors contributed equally as second authors: Jorge Oller, Sara Martínez-Martínez. [10]These authors contributed equally as senior authors: Miguel R Campanero, Juan Miguel Redondo. ✉E-mail: mcampanero@cbm.csic.es; jmredondo@cbm.csic.es

VCAN and ACAN have both been found to accumulate in the aortic medial layer of patients with diverse TAAD etiologies (Cikach et al, 2018). However, the role of VCAN and ACAN in MFS aortopathy has remained unknown.

VCAN participates in VSMC adhesion, migration, and proliferation (Evanko et al, 1999; Evanko et al, 2001) and important physiological and pathological processes, including cardiac development, angiogenesis, atherosclerosis, and bicuspid aortic valve formation (Asano et al, 2017; Dupuis et al, 2013; Mjaatvedt et al, 1998; Wight and Merrilees, 2004). Four VCAN isoforms (V0, V1, V2, V3) are produced as a result of alternative RNA splicing of exons encoding GAG attachment domains. While the V0 isoform contains the GAGα and GAGβ domains, V1 and V2 each contain only one of these domains (GAGβ and GAGα, respectively), and V3 has none. The V2 isoform is primarily found in the nervous system (Dours-Zimmermann et al, 2009), whereas the developing heart, VSMCs, and fibroblasts mainly express V0 and V1 (Evanko et al, 1999). Although ACAN has been mainly described in cartilage and the brain (Guilak et al, 2018; Morawski et al, 2012), it is emerging as an important contributor to vascular disease. ACAN is implicated in arterial remodeling in a pig model of intracoronary stenting (Suna et al, 2018), and altered ACAN cleavage correlates with aortic wall anomalies during development (Dupuis et al, 2013).

VCAN and ACAN are cleaved by ADAMTS1 (A disintegrin and metalloproteinase with thrombospondin motifs 1) and other members of the ADAMTS family (Mead and Apte, 2018; Sandy et al, 2001). We recently identified ADAMTS1 as an important mediator of vascular homeostasis and showed that ADAMTS1 expression is reduced in the aortic medial layer of MFS patients and a mouse model of MFS (Oller et al, 2017). The reduction in ADAMTS1 correlated with elevated expression of inducible nitric oxide synthase (NOS2) in the same samples (Oller et al, 2017); however, the mechanisms linking ADAMTS1 downregulation to NOS2 induction remained unknown.

Here, we investigated if aortopathy in MFS is driven by the accumulation of the uncleaved forms of the ADAMTS1 substrates VCAN and ACAN. We show that VCAN accumulates in the aortas of MFS patients and the $Fbn1^{C1041G/+}$ mouse model of MFS and demonstrate that Vcan accumulation mediates Nos2 induction and aortopathy through Akt activation.

# Results

## Vcan accumulates in the aortic wall of MFS mice

Since ADAMTS1 cleaves VCAN and ACAN in the aorta (Fava et al, 2018; Sandy et al, 2001) and is downregulated in the aorta of MFS patients and mice (Oller et al, 2017), we investigated whether both substrates accumulate in this tissue in MFS. Vcan and Acan cleavage by Adamts1 generates the neoepitopes neo-Vcan and neo-Acan, which can be detected with specific antibodies. Consistent with the decreased Adamts1 expression, aortic tissue from 12-week-old MFS mice expressed lower levels of neo-Vcan and increased levels of Vcan (Fig. 1A–C). However, there were no discernable changes in the content of aortic Vcan content in 4-week-old mice (Fig. EV1), suggesting that Vcan accumulation in MFS mouse aorta occurs somewhere between 4 and 12 weeks of

age. Although neo-Acan levels were also depressed, uncleaved Acan did not accumulate (Fig. 1D–F). VCAN and ACAN immunostaining in aortic sections from organ transplant donors and MFS patients revealed accumulation of both PGs in the aortic medial layer of MFS patients (Figs. 1G–I and EV2A,B), regardless of sex or age.

## Vcan haploinsufficiency protects MFS mice from aortic enlargement

To investigate the contribution of Vcan accumulation to MFS aortic disease, we used Vcan-deficient mice. Homozygous $Vcan^{hdf/hdf}$ mice die during embryonic development due to heart defects (Mjaatvedt et al, 1998), so we used haploinsufficient mice ($Vcan^{hdf/+}$), a mouse model with reduced Vcan protein expression (Asano et al, 2017). To characterize the effect of Vcan haploinsufficiency on the aortic phenotype of $Fbn1^{C1041G/+}$ MFS mice, we performed ultrasound imaging at different ages (12, 24, and 36 weeks). This analysis revealed that Vcan haploinsufficiency conferred protection against dilation of the ascending aorta (AsAo) and abdominal aorta (AbAo) in MFS mice at all analyzed ages, with $Fbn1^{C1041G/+};Vcan^{hdf/+}$ (MFS $Vcan^{hdf/+}$) mice exhibiting only a slightly larger aortic diameter than that found in WT mice (Fig. 2A–D).

## Acan does not contribute to MFS aortopathy

We similarly investigated the potential contribution of Acan to MFS aortopathy, using $Acan^{cmd/+}$ haploinsufficient mice ($Acan^{cmd/cmd}$ mice die just after birth due to respiratory failure (Watanabe et al, 1997)). Ultrasound measures of AsAo and AbAo diameters in 12, 24, and 36-week-old mice showed no effect of Acan haploinsufficiency on AsAo or AbAo dilatation at any age tested (Fig. 3A–C).

As an alternative approach to assess the role of Acan in aortic disease in adult MFS mice, we knocked down Acan expression in vivo in aortas of 12-week-old MFS mice by intrajugular vein injection of lentiviral vectors (Alfranca et al, 2018; Esteban et al, 2011) encoding an Acan-specific shRNA and green fluorescent protein (GFP) to monitor transduction efficiency. Lentiviral knockdown efficiency was examined by transducing VSMCs from WT and MFS mice with shAcan-encoding lentiviruses or control lentiviruses containing a non-specific target sequence (shScr). Acan mRNA content was markedly reduced in VSMCs from WT and MFS aortas transduced with shAcan-expressing lentivirus (Fig. 4A). After intrajugular delivery of these lentiviruses to 12-week-old WT and MFS mice, we monitored aortic diameter overtime (Fig. 4B). Notable GFP immunostaining was evident 28 days after lentiviral administration in AsAo and AbAo sections from infected mice (Fig. 4C), and Acan protein expression was efficiently knocked down only in aortic samples from mice inoculated with shAcan-expressing lentivirus (Fig. 4D,E). Consistent with the data from the $Acan^{cmd/+}$ mouse model, aortic Acan silencing failed to regress aortic dilation in MFS mice (Fig. 4F). In addition to aortic enlargement, Fbn1 mutations cause medial degeneration, characterized by elastic fiber fragmentation and disarray, and medial thickening (Romaniello et al, 2014). Histological analysis 28 days after lentiviral delivery revealed that shAcan-transduced MFS aortas showed no improvement in medial degeneration relative to shScr-transduced MFS aortas (Fig. 5A–C).

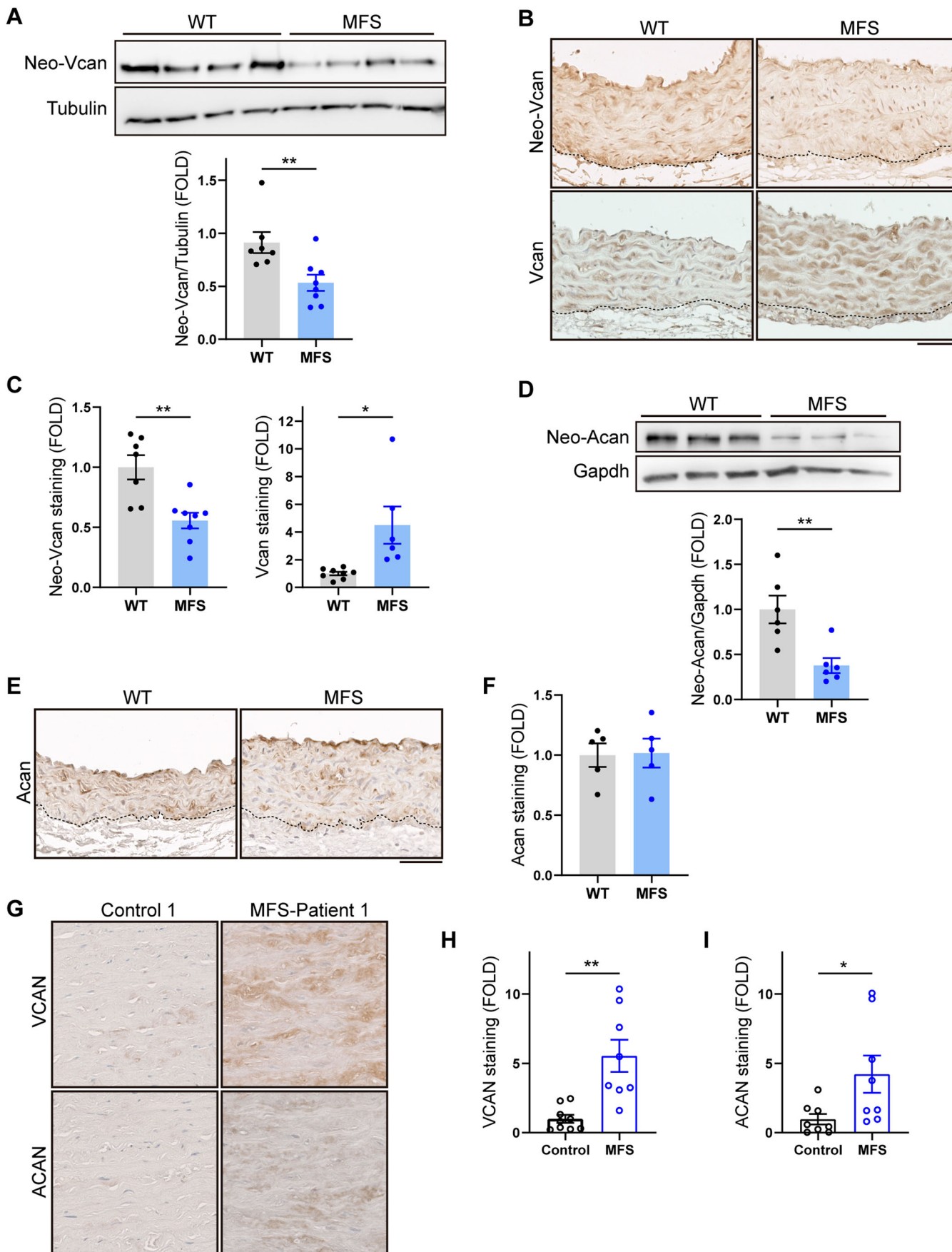

**Figure 1. Vcan accumulates in the aortas of *Fbn1^C1041G/+* MFS mice.**

(A) Representative immunoblot analysis of neo-Vcan in aortic extracts from WT and MFS mice, with quantification of the immunoblot signal (*n* = 7–8 mice per group). (B) Representative immunohistochemistry images of neo-Vcan and Vcan in mouse AsAo. Dotted lines outline the boundaries between the media and the adventitia. Scale bar, 50 μm. (C) Quantification of neo-Vcan and Vcan immunostaining in aortic sections from WT and MFS mice (*n* = 6–8 mice). (D) Representative immunoblot analysis of neo-Acan in aortic extracts from WT and MFS mice, with quantification of the immunoblot signal (*n* = 6 mice per group). (E) Representative images of Acan immunohistochemistry in mouse AsAo. Dotted lines outline the boundaries between the media and the adventitia. Scale bar, 50 μm. (F) Quantification of Acan immunostaining in AsAo from WT (*n* = 5) and MFS (*n* = 5) mice. (G) Representative medial layer images of VCAN and ACAN immunohistochemistry in aortic cross sections of human samples from control donors and MFS patients. Scale bar, 50 μm. (H, I) Quantification of VCAN (H) and ACAN (I) immunohistochemistry in aortas from control donors and MFS patients (*n* = 8–9 samples per group). Data information: (A, C, D, F) data are shown relative to WT mice as mean ± s.e.m. Each data point denotes an individual mouse. *P < 0.05, **P < 0.01 (Student *t* test or unpaired *t* test with Welch's correction, as appropriate). (H, I) Data are shown relative to control donors as mean ± s.e.m. Each data point denotes an individual. *P < 0.05, **P < 0.01 (unpaired *t* test with Welch's correction). Source data are available online for this figure.

## Vcan accumulation mediates aortopathy in MFS and is required for Nos2 overexpression

To confirm the beneficial effect of *Vcan* deficiency on aortic dilation in adult MFS mice, we knocked down aortic Vcan expression in vivo. After confirming the silencing capacity of shVcan-encoding lentiviruses in cultured VSMCs obtained from WT and MFS aortas (Fig. 6A), we inoculated mice with these viruses by intrajugular injection and analyzed their effects on aortic diameter (Fig. 6B). Intrajugular delivery of lentivirus encoding shScr or shVcan yielded efficient transduction of all aortic layers, as determined by GFP immunostaining of AsAo and AbAo sections at 4 weeks post-injection (Fig. 6C). In MFS mice transduced with shVcan, this was concomitant with an effective knockdown of *Vcan* in aortic samples that was not observed in WT aortas (Fig. 6D,E). This disparity is likely due to a slower turnover rate of Vcan in WT aortas. *Vcan* silencing in the aortas of MFS mice completely reversed AsAo and AbAo enlargement 14–21 days after lentiviral transduction (Fig. 6F).

This reduction in aortic diameter was accompanied by a recovery of tunica media structure and architecture. Histological analysis of aortic cross sections 28 days after lentiviral administration showed a marked reversal of medial degeneration, evidenced by regression of elastic fiber fragmentation and disarray (Fig. 7A,B), and of aortic wall thickening (Fig. 7A,C).

The aortas of MFS mice and patients overexpress NOS2, which mediates aortopathy in this setting (Oller et al, 2017). We therefore determined Nos2 protein content in the aortic wall of MFS mice transduced with the shVcan lentivirus. Aortic *Vcan* silencing reduced Nos2 expression to normal levels, whereas shScr-encoding lentivirus had no effect (Figs. 7D,E and EV3A). Increased Nos2 protein levels in MFS aortas were identified primarily in smooth muscle cells (Sma-positive cells), but the possibility of Nos2 expression by some endothelial cells cannot be entirely ruled out (Fig. EV3B,C). Together, these findings suggest that *Vcan* knockdown reverses MFS aortopathy by decreasing Nos2 expression.

## Vcan mediates Nos2 upregulation and MFS aortic disease via Akt activation

We next considered the possibility that Vcan accumulation contributes to MFS aortopathy via activation of the Akt signaling pathway, a hypothesis based on three observations: (i) *Adamts1* knockdown activates Akt (Oller et al, 2017), a well-known mediator

of Nos2 transcriptional activation (Tang et al, 2007); (ii) pharmacological inhibition of Akt regresses aortic dilation in *Adamts1* deficient mice (Oller et al, 2017); and (iii) *Vcan* knockdown significantly reduced Nos2 protein levels in MFS mouse aorta. Supporting this idea, plating of WT VSMCs on VCAN V3-coated plates activated Akt, as determined by immunoblot analysis of Akt Ser473 phosphorylation (Fig. 8A). Furthermore, pharmacological inhibition of Akt signaling with the mTOR inhibitor AZD8055 in MFS mice (Fig. 8B) decreased AsAo and AbAo dilation to normal levels within 4 days of treatment (Fig. 8C).

Since Nos2 is induced by Akt activation (Tang et al, 2007), we investigated whether Nos2 overexpression in MFS aorta was dependent on mTOR/Akt signaling. Remarkably, mTOR/Akt inhibition substantially reduced aortic Nos2 protein levels in MFS mice (Figs. 8D,E and EV4). We further assessed the activation status of the AKT pathway in MFS patients, finding higher levels of pAKT-S473 in aortas from MFS patients than in aortic sections from control organ transplant donors (Figs. 8F and EV5), regardless of sex or age. Quantification of the pAKT-S473-positive area in stained sections confirmed significantly higher AKT phosphorylation in aortas from MFS patients (Fig. 8G). Taken together, these findings show that Vcan accumulation mediates aortic Nos2 induction through Akt activation in MFS.

# Discussion

PG accumulation is a hallmark of medial degeneration in TAAD (Shen et al, 2019). Here, we have demonstrated that VCAN accumulates in the aortas of *Fbn1^C1041G/+* mice and MFS patients and mediates aortopathy. Our findings further show that Vcan activates Akt and identify the AKT signaling pathway as a potential target for therapeutic intervention in human MFS aortic disease.

In an earlier study, we showed that *Adamts1* deficiency causes an aortic syndromic disease similar to that found in MFS mice, identified Adamts1 as an important mediator of vascular homeostasis, and demonstrated depressed levels of aortic ADAMTS1 protein in MFS mice and patients (Oller et al, 2017). However, the role in MFS aortopathy played by VCAN and ACAN (the main substrates of ADAMTS1 in aorta) remained unknown. Our present results indicate that Vcan, but not Acan, accumulates in the aortas of *Fbn1^C1041G/+* MFS mice, likely as a result of reduced Adamts1 protein expression and decreased Vcan cleavage.

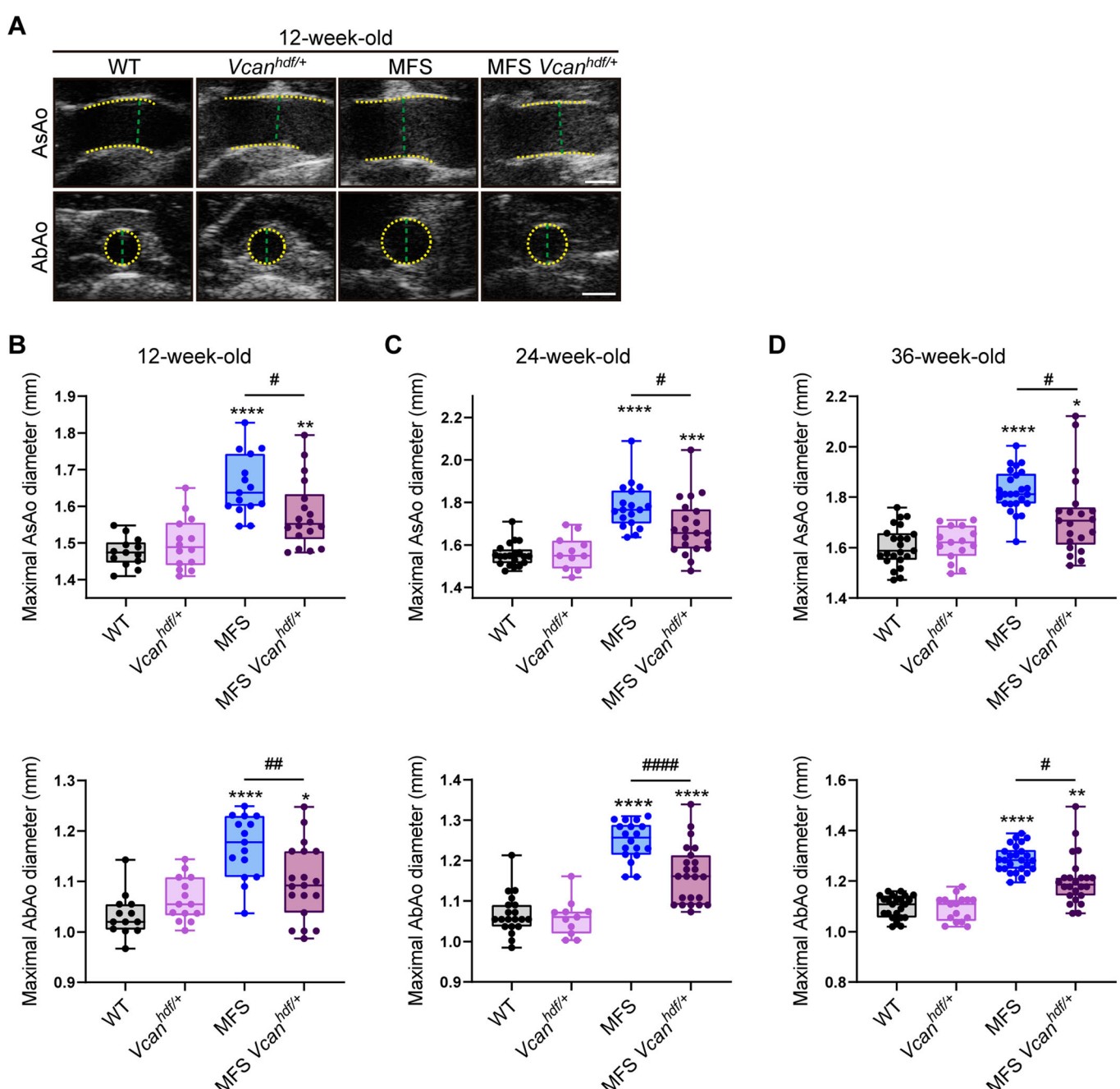

**Figure 2.  *Vcan* haploinsufficiency protects MFS mice from aortic dilation.**

(A) Representative in vivo ultrasound images of the AsAo and AbAo from 12-week-old WT, *Vcan*<sup>hfd/+</sup>, MFS, and MFS *Vcan*<sup>hdf/+</sup> mice. Yellow dashed lines delineate lumen boundaries and green dashed lines denote lumen diameters. Scale bar, 1 mm. (B–D) Maximal AsAo and AbAo diameters in *Vcan* haploinsufficient mice aged (B) 12, (C) 24, and (D) 36 weeks. Data information: (B–D) data are shown as box-and-whisker plots. The box itself spans from the first quartile (25%) to the third quartile (75%), representing the interquartile range where the central 50% of data values fall. Inside the box, a line denotes the median value. The whiskers of the boxplot extend from the ends of the box to the minimum and maximum values. Each data point denotes an individual mouse (12-week-old mice: WT, $n = 13$; *Vcan*<sup>hdf/+</sup>, $n = 14$; MFS, $n = 15$; MFS;*Vcan*<sup>hdf/+</sup>, $n = 18$. 24-week-old mice: WT, $n = 20$; *Vcan*<sup>hdf/+</sup>, $n = 11$; MFS, $n = 18$; MFS;*Vcan*<sup>hdf/+</sup>, $n = 21$. 36-week-old mice: WT, $n = 23$; *Vcan*<sup>hdf/+</sup>, $n = 15$; MFS, $n = 25$; MFS;*Vcan*<sup>hdf/+</sup>, $n = 21$). $*P < 0.05$, $**P < 0.01$, $***P < 0.001$, $****P < 0.0001$ versus WT mice; $^{\#}P < 0.05$, $^{\#\#}P < 0.01$, $^{\#\#\#\#}P < 0.0001$ versus MFS mice (two-way ANOVA with Tukey's post hoc test or the Kruskal–Wallis test with Dunn's multiple comparison test, as appropriate). Source data are available online for this figure.

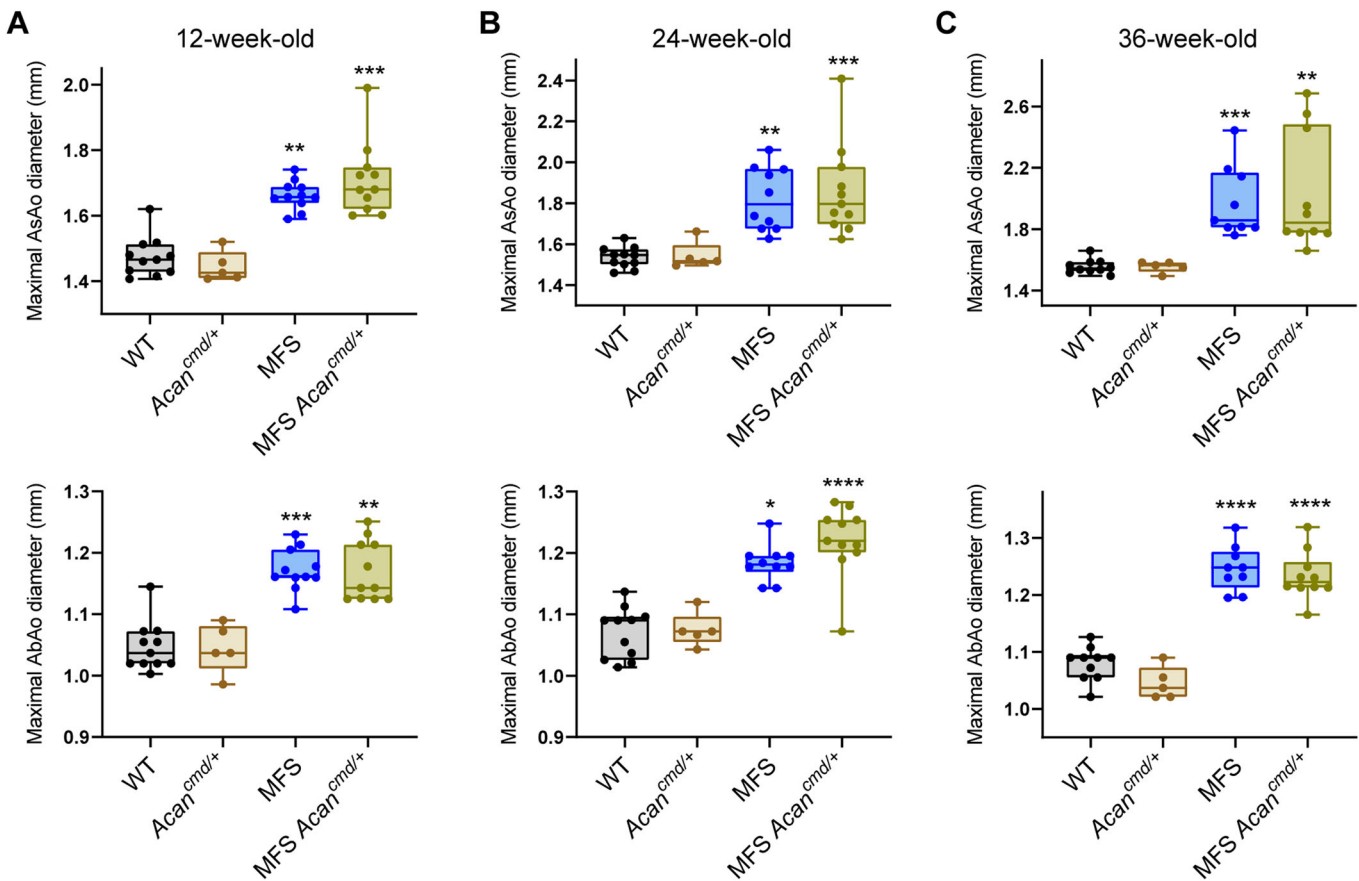

**Figure 3. Acan haploinsufficiency does not prevent aortic enlargement in MFS mice.**

(A–C) Maximal AsAo and AbAo diameters in (A) 12-, (B) 24-, and (C) 36-week-old mice (WT, n = 10–11; Acan^cmd/+, n = 5; MFS, n = 9–11; MFS;Acan^cmd/+, n = 10–11). Data information: (A–C) data are shown as box-and-whisker plots. The box itself spans from the first quartile (25%) to the third quartile (75%), representing the interquartile range where the central 50% of data values fall. Inside the box, a line denotes the median value. The whiskers of the boxplot extend from the ends of the box to the minimum and maximum values. Each data point denotes an individual mouse. *P < 0.05, **P < 0.01, ***P < 0.001, ****P < 0.0001 versus WT mice (two-way ANOVA with Tukey's post hoc test or the Kruskal–Wallis test with Dunn's multiple comparison test, as appropriate). Source data are available online for this figure.

Consistent with decreased Adamts1 protein expression, neo-Vcan and neo-Acan are decreased in MFS aortas. However, while total Vcan protein accumulates in the aorta of $Fbn1^{C1041G/+}$ mice, total aortic Acan protein content is similar to that in WT littermates, consistent with evidence that proteins other than Adamts1 may contribute to Acan degradation (Lark et al, 1997). Indeed, other Adamts family members with aggrecanase activity could also contribute to the aortic disease in MFS. For instance, Adamts5 has been implicated in TAA, as mice lacking the catalytic domain of Adamts5 exhibit a larger aortic dilatation following AngII treatment than AngII-treated WT mice (Fava et al, 2018).

Intriguingly, previous studies showed that Acan accumulates in the aorta of hypomorphic $Fbn1^{mgR/mgR}$ MFS mice, particularly after AD or rupture (Cikach et al, 2018). It is worth noting, however, that whereas the $Fbn1^{mgR/mgR}$ mouse model replicates a severe form of MFS that leads to lethal aortic rupture in almost 50% of mice during the first three months of life (Galatioto et al, 2018; Pereira et al, 1999), the $Fbn1^{C1041G/+}$ mouse model recapitulates a milder

aortic disease that rarely progresses to AD (Judge et al, 2004). Increased levels of VCAN and ACAN have been reported in the aortas of TAAD patients, including a single patient with MFS (Cikach et al, 2018). Consistent with these results, the content of hyaluronan, a widely distributed GAG in the ECM that binds to VCAN and ACAN, is increased in the tunica media of MFS patients and in cultured MFS-derived VSMCs (Nataatmadja et al, 2006). Moreover, a glycoproteomic analysis showed that glycosylation of aortic Vcan and Acan is enhanced in MFS patients relative to non-MFS aneurysmal patients (Yin et al, 2019).

Our clinical data reveal VCAN accumulation in the aortic medial layer of almost every MFS patient, and although ACAN accumulation was less prevalent, it was detected in almost half of these patients. The lack of Acan aortic accumulation in $Fbn1^{C1041G/+}$ mice thus contrasts with the accumulation observed in $Fbn1^{mgR/mgR}$ mice and MFS patients. It may be that distinct $FBN1$ mutations (each with its specific penetrance) have different impacts on ACAN turnover. Alternatively, ACAN accumulation might be caused by

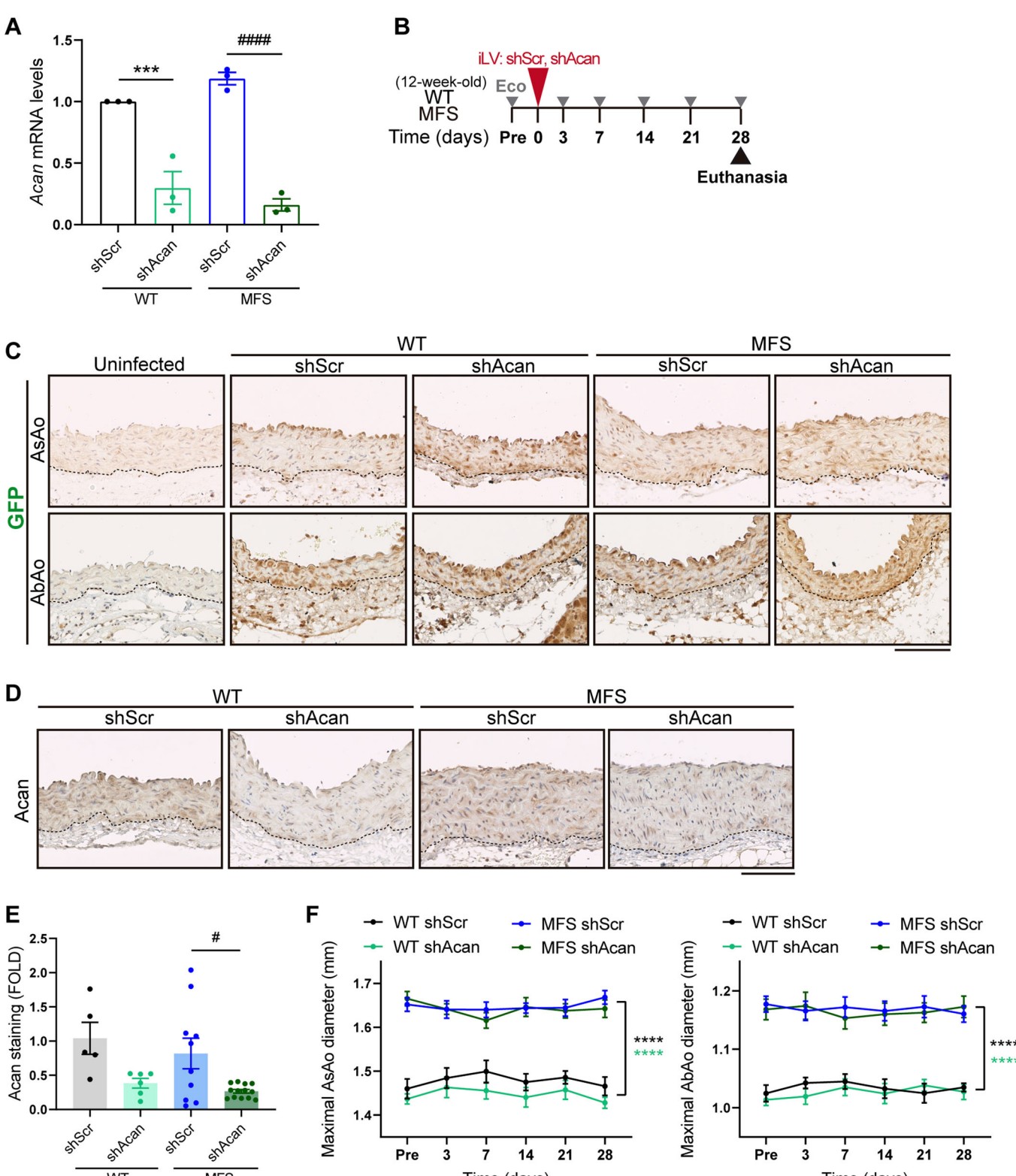

vascular wall deterioration at advanced stages of aortic disease or by an additional factor that contributes to a more severe aortic phenotype and AD. Consistent with this hypothesis, patients with acute type A AD have higher plasma ACAN than patients with TAA or healthy volunteers (König et al, 2021), supporting the possible use of ACAN as a prognostic biomarker for AD. Given the absence of predictive markers of AD, the identification of prognostic plasma markers is of the utmost importance, and it would be interesting to assess the prognostic value of plasma VCAN and ACAN for stratifying the risk of AD and rupture. To

◄ **Figure 4.** *Acan* **silencing does not reduce aortic dilation in MFS mice.**

(**A**) *Acan* mRNA expression assessed by RT-qPCR in WT and MFS VSMCs transduced with lentivirus encoding shScr- or *Acan*-specific shRNA (n = 3 biological replicates). (**B**) Experimental design: 12-week-old WT and MFS mice were inoculated with lentivirus (iLV) expressing shScr or shAcan, monitored for aortic dilation (Eco), and euthanized at 28 days. (**C**) Representative images of GFP immunostaining in AsAo and AbAo of uninfected mice or mice expressing shScr or shAcan. Dotted lines outline the boundaries between the media and the adventitia. Scale bar, 100 μm. (**D**) Representative images of Acan immunohistochemistry in AsAo from shScr- or shAcan-transduced WT and MFS mice. Dotted lines outline the boundaries between the media and the adventitia. Scale bar, 100 μm. (**E**) Quantification of Acan immunostaining in aortic sections from the mouse cohorts shown in D (WT shScr, n = 5; WT shAcan, n = 6; MFS shScr, n = 10; MFS shAcan, n = 13 mice per group). (**F**) Maximal AsAo and AbAo diameters in WT shScr (n = 7), WT shAcan (n = 8), MFS shScr (n = 14), and MFS shAcan (n = 17) mice at the indicated time points. Data information: (**A**) \*\*\*P < 0.001 versus shScr-infected WT VSMCs; ####P < 0.0001 versus shScr-infected MFS VSMCs (two-way ANOVA with Tukey's post hoc test). (**E**) Data are shown relative to shScr-transduced WT mice as mean ± s.e.m. Each data point denotes an individual mouse. #P < 0.05 versus shScr-transduced MFS mice (two-way ANOVA with Tukey's post hoc test). (**F**) Data are shown as mean ± s.e.m. \*\*\*\*P < 0.0001 versus shScr-transduced MFS mice (repeated-measurements two-way ANOVA with Tukey's post hoc test). Source data are available online for this figure.

explore this avenue, we measured plasma Vcan levels in our mouse model of MFS and observed a modest increase in mice at 36–40 weeks of age compared to age-matched WT mice (Appendix Fig. S1). Since *Fbn1^{C1041G/+}* mice are not predisposed to AD, it would be valuable to assess plasma Vcan levels in other mouse models of MFS that exhibit more severe aortic pathology, as well as in patients with advanced development of aortic pathology. Such studies could provide insights into the clinical relevance of Vcan as a potential biomarker for AD risk stratification.

Nevertheless, our data show that *Acan* haploinsufficiency or knockdown failed to rescue aortic dilation or any feature of MFS-related aortopathy in the *Fbn1^{C1041G/+}* mouse model. These results suggest that *Acan* deficiency is not sufficient to reverse aortic pathology, ruling out Acan as a causative contributor to MFS aortopathy, at least at the disease stages analyzed. In contrast, the reversal of aortic dilation and medial degeneration upon 28 days of *Vcan* silencing and the partial protection of *Vcan* haploinsufficient mice against aortic enlargement together provide strong evidence that Vcan mediates aortopathy in MFS.

Modulation of VCAN content has profound effects on cell behavior. For example, VCAN accumulation impairs focal adhesions in human uterine smooth muscle cells, predisposing cells to an anti-adhesive state (Mead et al, 2018). Moreover, VCAN accumulation leads to a myofibroblast-like phenotype by promoting ACTA2 expression in fibroblasts and thus increasing their contractility (Carthy et al, 2015; Hattori et al, 2011). Through its interaction with hyaluronan, VCAN also binds to other ECM components and cell-surface protein receptors, including CD44. Our results suggesting a deleterious effect of Vcan accumulation are in line with data showing that *CD44^{-/-}* mice have a lower incidence of TAAD than WT mice in response to the simultaneous infusion of AngII and the lysyl oxidase inhibitor β-aminopropionitrile (BAPN)(Hatipoglu et al, 2020).

Nos2 plays a paramount role in MFS aortopathy (Nettersheim et al, 2022; Oller et al, 2017; Toral et al, 2022). MFS is associated with increased plasma levels of cGMP and nitrated proteins, elevated levels of circulating and tissue nitric oxide (NO), and activation of PKG in aorta, indicating NO signaling pathway overactivation in this aortic disease (de la Fuente-Alonso et al, 2021; Oller et al, 2017). Our data show that *Vcan* silencing reduces Nos2 expression in the aortas of MFS mice, and this could by itself account for the regression of MFS aortopathy. Nos2 reduction by *Vcan* silencing would decrease soluble Guanylate Cyclase (sGC) activation, thus reducing cGMP production and the subsequent activation of PKG, all signaling components of the NO pathway

shown to mediate MFS aortopathy in mice (de la Fuente-Alonso et al, 2021). In line with these results, nitro-oleic acid has been shown to inhibit Nos2 expression and ameliorate aortic dilation in MFS mice (Nettersheim et al, 2022).

Our results show that Vcan activates Akt and identify Akt as an important contributor to MFS aortopathy. These findings are consistent with others showing that VCAN activates AKT signaling in hepatocellular carcinoma and breast cancer cells (Du et al, 2013; Zhangyuan et al, 2020). Also consistent with our results, a recent single-cell RNA sequencing analysis showed that high *VCAN* mRNA levels correlated with enrichment in genes associated with PI3K-AKT signaling in aortic cells from patients with sporadic TAA (Song et al, 2023). In this regard, VCAN interaction with protein receptors at the cellular surface, including EGFR, CD44 or the β1 integrin (Wu et al, 2005) might activate AKT and consequently induce NOS2 expression. Our findings indicating increased AKT activation in aortas from MFS patients align with results published after submission of this manuscript showing heightened Akt signaling in *Fbn1^{C1041G/+}* mouse aortas (Nakamura et al, 2023).

Supporting a causal role for Akt activation in MFS aortopathy, our data indicate that pharmacological mTOR/Akt inhibition rapidly reverses aortic dilation and decreases Nos2 levels in MFS mice. In line with the regression of aortic dilatation in such a short period of time, pharmacological inhibition of sGC or PKG1 regressed aortic dilation in MFS mice after just 1 week of treatment (de la Fuente-Alonso et al, 2021). It should be noted, however, that the regression of medial degeneration required longer periods of time (de la Fuente-Alonso et al, 2021). The mTOR pathway has previously been linked to aneurysmal progression, with chronic hyperactivation of mTORC1 leading to progressive medial degeneration and TAAD by inducing a degradative aortic VSMC phenotype (Li et al, 2020). Accordingly, the mTORC1 inhibitor rapamycin has been proposed as a potential medical treatment for TAAD because it confers protection against TAAD in MFS mouse models (Liu et al, 2019). Of note, Rictor, a member of the mTORC2 complex responsible for Akt activation, is overactivated in the aortas of aged MFS mice (Parker et al, 2018). The insulin signaling pathway may also contribute to MFS pathology through AKT, as insulin sequestration by fibrillin-1 attenuates the PI3K/AKT/mTORC signaling pathway (Muthu et al, 2022). Moreover, a DNA-methylation analysis conducted on peripheral blood samples from MFS patients revealed differential methylation of insulin-associated genes significantly associated with MFS aortopathy (van Andel et al, 2021). These studies are consistent with our data,

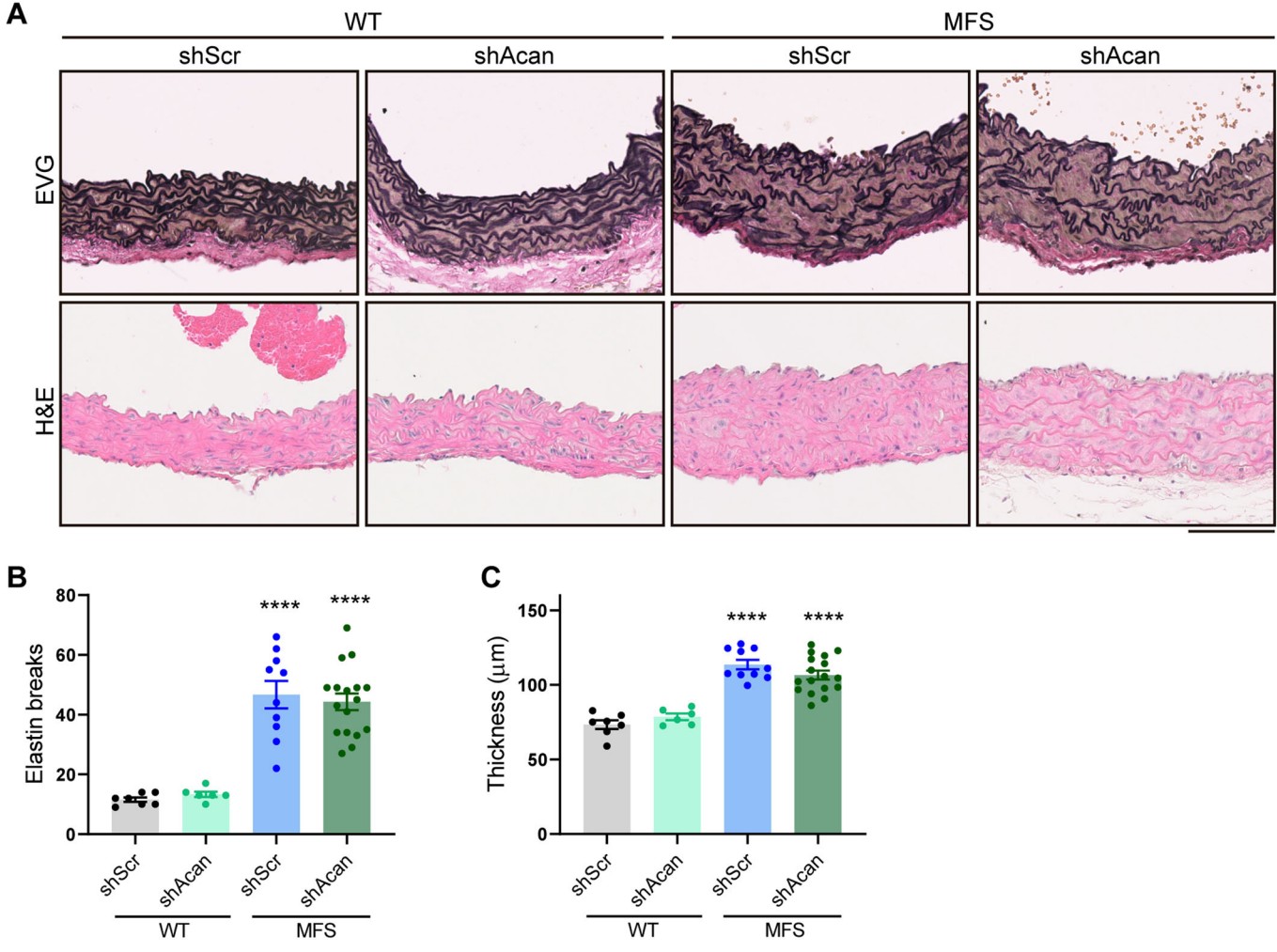

**Figure 5.** *Acan* **silencing does not regress medial degeneration in MFS mice.**

(A) Representative images showing elastic van Gieson (EVG) and hematoxylin and eosin (H&E) staining in AsAo of the indicated mice ($n = 6$–17 mice per group). Scale bar, 100 μm. (B, C) Quantification of elastin breaks (B) and medial wall thickness (C) in AsAo from the mouse cohorts shown in (A). Data information: (B, C) data are shown as mean ± s.e.m. Each data point denotes an individual mouse. ****$P < 0.0001$ versus shScr- transduced WT mice (two-way ANOVA with Tukey's post hoc test). Source data are available online for this figure.

which underscore the pivotal role of the mTOR/Akt pathway in MFS aortic disease. Of note, the increased AKT activation observed in aortas from MFS patients does not appear to correlate with elastic fiber fragmentation and disarray. However, further investigations are warranted to determine whether such activation is linked to aortic enlargement.

Taken together, our findings indicate that activation of the Akt pathway by Vcan accumulation leads to Nos2 overexpression and increased Nos2-derived NO production, triggering overactivation of the NO-sGC-PKG signaling pathway and thus promoting medial degeneration and aortic disease (Fig. 9). Since our results also reveal elevated levels of phosphorylated AKT in the aortas of MFS patients, we urge consideration of the mTOR-Akt pathway as a potential therapeutic target for MFS aortopathy.

# Methods

## Animal procedures

Animal procedures and experiments complied with all relevant ethical regulations, were accredited by the CNIC Ethics Committee and the Madrid regional authorities (ref. PROEX 80/ 16 and PROEX 094.8/21), and conformed to EU Directive 2010/63/ EU and Recommendation 2007/526/EC regarding the protection of animals used for experimental and other scientific purposes, enacted in Spanish law under Real Decreto 1201/2005. Mouse health was daily assessed for signs of discomfort; weight loss; or changes in behavior, mobility and feeding, or drinking habits. Mice were housed in a pathogen-free animal facility under a 12 h light/dark cycle at constant temperature and

     

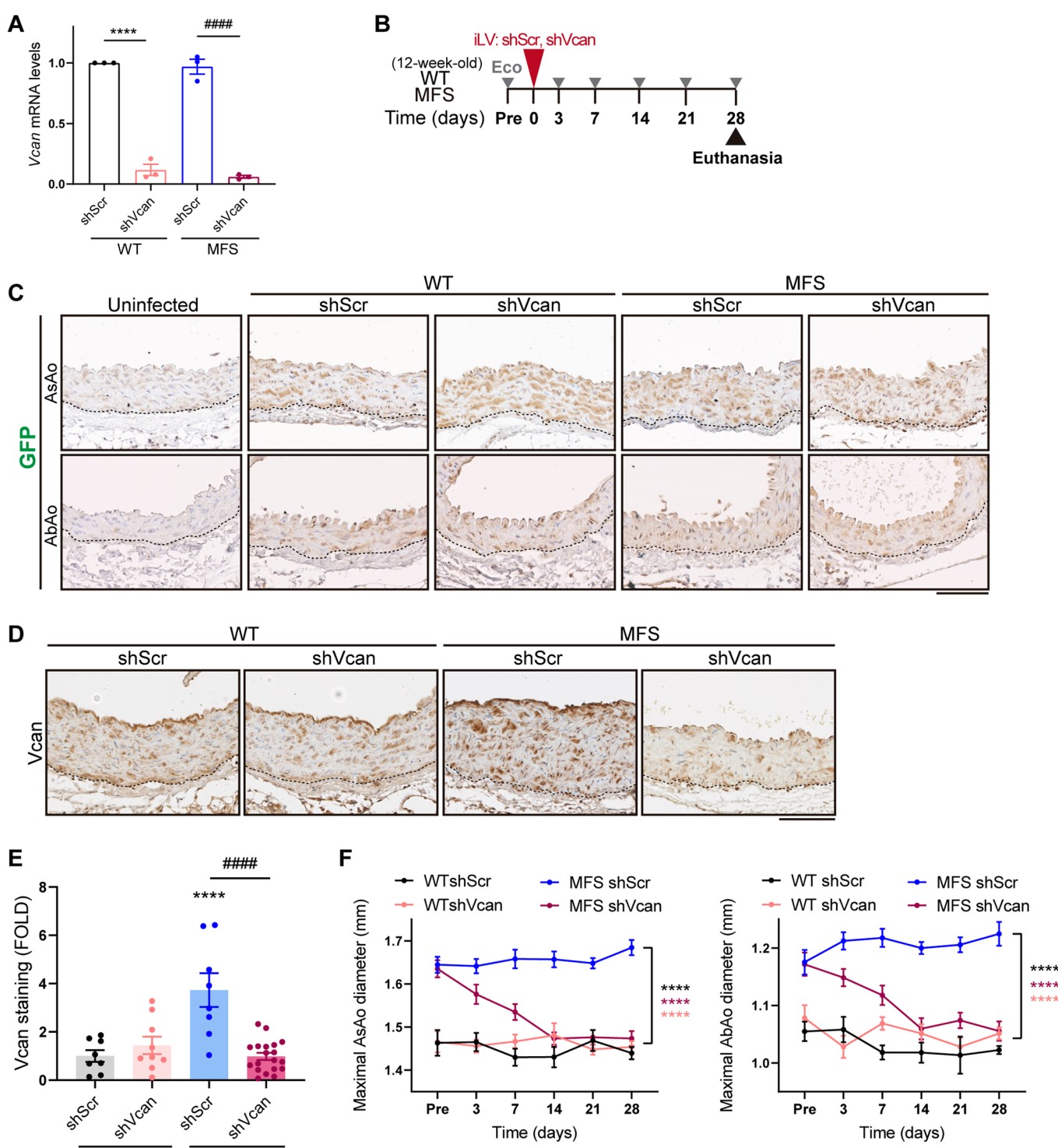

humidity and had access to standard rodent chow and water ad libitum.

*Fbn1^C1041G/+* MFS mice (Judge et al, 2004), which harbor a mutation in the *Fbn1* gene, and *Acan^cmd/+* mice, containing a 7 pb-deletion in exon 5 of the *Acan* gene (Rittenhouse et al, 1978), were obtained from Jackson Laboratories (JAX mice stock #012885 and #010522, respectively). *Vcan^hdf/+* mice carry a β-galactosidase (*LacZ*)

cassette between exons 7 and 8 of the genomic sequence of the *Vcan* gene (Mjaatvedt et al, 1998). All strains had been previously backcrossed to the C57BL/6 background for more than ten generations. All mice were genotyped by PCR of tail samples using the following primers: *Fbn1^C1041G/+* mice (5′-CTCATCATTTTTGGC-CAGTTG-3′, 5′-GCACTTGATGCACATTCACA-3′); *Vcan^hdf/+* mice (5′-CGGCCAGGACAGTCGTTTGCCGTCTG-3′, 5′-CCTGACCATG

**Figure 6.  Vcan silencing regresses aortic dilation in MFS mice.**

(A) *Vcan* mRNA expression assessed by RT-qPCR in WT and MFS VSMCs transduced with lentivirus encoding shScr or *Vcan*-specific shRNA (n = 3 biological replicates). (B) Experimental design: 12-week-old WT and MFS mice were inoculated with lentivirus (iLV) expressing shScr or shVcan, monitored for aortic dilation (Eco), and euthanized at 28 days. (C) Representative images of GFP immunostaining in AsAo and AbAo of uninfected mice of mice expressing shScr- or shVcan. Dotted lines outline the boundaries between the media and the adventitia. Scale bar, 100 µm. (D) Representative images of Vcan immunohistochemistry in AsAo from shScr- or shVcan-transduced WT and MFS mice. Dotted lines outline the boundaries between the media and the adventitia. Scale bar, 100 µm. (E) Quantification of Vcan immunostaining in aortic sections from the mouse groups shown in (D) (WT shScr, n = 8; MFS shVcan, n = 9; MFS shScr, n = 8; and MFS sh*Vcan*, n = 20 mice per group). (F) Maximal AsAo and AbAo diameters at the indicated time points in WT shScr (n = 6), WT shVcan (n = 9), MFS shScr (n = 13), and MFS shVcan (n = 15) mice. Data information: (A) ****P < 0.0001 versus shScr-infected WT VSMCs; ####P < 0.0001 versus shScr-infected MFS VSMCs (two-way ANOVA with Tukey's post hoc test). (E) Data are shown relative to shScr-transduced WT mice as mean ± s.e.m. Each data point denotes an individual mouse. ****P < 0.0001 versus shScr-transduced WT mice; ####P < 0.0001 versus shScr-transduced MFS mice (two-way ANOVA with Tukey's post hoc test). (F) Data are shown as mean ± s.e.m. ****P < 0.0001 versus shScr-transduced MFS mice (repeated-measurements two-way ANOVA with Tukey's post hoc test). Source data are available online for this figure.

CAGAGGATGATGCTCG-3'); and *Acan*$^{cmd/+}$ mice (5'-CCGGGA CACCAATGAGACCTATG-3', 5'-GGAATCCGGGACACCAATGA GATG-3', 5'-GCATGGATTCCAGCAAGAGACCA-3'). Wild-type (WT) littermates were used as controls unless otherwise specified. The mTOR inhibitor AZD8055 (MedChem Express, HY-10422) was dissolved in 8% DMSO in corn oil and administered intraperitoneally at 20 mg/kg/day for 4 days. Control mice were inoculated intraperitoneally with vehicle solution (Veh).

## In vivo ultrasound imaging

Aortic diameter was monitored in isoflurane-anesthetized mice (2% isoflurane) by high-frequency ultrasound with a VEVO 2100 echography device (VisualSonics, Toronto, Canada) fitted with a 30-micron axial resolution transducer. Maximal internal aortic diameter was measured at systole using VEVO 2100 software, version 3.2.0. Measurements were taken before treatment initiation to determine baseline aortic dimensions and at the indicated time points during the experiment.

## Cell procedures

Primary mouse vascular smooth muscle cells (VSMCs) were obtained from 8–10-week-old male WT and MFS mice as previously described (Villahoz et al, 2018). Briefly, adventitia-free aortas were digested with 1 mg/ml collagenase type II and 0.5 mg/ml elastase (Worthington, LS004176 and LS002292, respectively) until a single-cell suspension was obtained. Cells were then cultured at 37 °C, 5% $CO_2$ in Dulbecco's modified Eagle's medium (DMEM) supplemented with 20% fetal bovine serum (FBS), 2 mM L-glutamine, 100 U/ml penicillin, and 100 µg/ml streptomycin. All experiments were performed in passage 2–6.

For coating experiments, plates were first coated overnight at 4 °C with 20 µg/ml recombinant human VCAN V3 (R&D, 3054-VN-050) or PBS as control and blocked with 0.25% BSA for 30 min at 37 °C. After seeding serum-starved VSMCs at 90% confluence on the PBS- or V3-precoated surface, the plates were centrifuged for 5 min at 200×g and 4 °C to ensure adherence. After incubation for 1 h at 37 °C, VSMCs were harvested for protein assays.

HEK-293T (CRL-11268) and Jurkat (TIB-152, clone E61) cell lines, purchased from ATCC, were used for high-titer lentiviral production and titration, respectively, and tested negative for *Mycoplasma*. These cell lines were not authenticated for this project; instead, cells were used after receipt or after resuscitation from early stocks at low passage numbers.

## Lentivirus production and infection

Lentiviral plasmids encoding GFP and a shRNA targeting mouse *Vcan* or *Acan* were engineered by cloning into the pH1-DUAL lentiviral vector the following shRNA sequences: shAcan (ACAA-CAGAAGTGCCATATTTC); shVcan (AGCTAGTCCGGAGATT-GATAA). Non-specific shRNA (shScr, CAACAAGATGAAGAG CACCAA) was used as a control.

Pseudo-typed lentiviruses were produced by transient calcium phosphate transfection of HEK-293T. Following medium replacement, cells were harvested after 48 h, and the supernatant was ultracentrifuged for 2 h at 26,000 r.p.m. (SW28 rotor, Optima L-100 XP Ultracentrifuge, Beckman). Lentiviruses were resuspended in cold PBS and stored at −80 °C to avoid degradation. Lentiviral vectors were titrated in Jurkat cells. Briefly, Jurkat cells were seeded on a 96-well plate and infected with lentiviral dilutions from 1/10 to 1/10,000. After 48 h, cells were centrifuged for 5 min at 200×g and 4 °C and resuspended in cold PBS containing propidium iodide (Sigma, P4864, 1/1000). Transduction efficiency (GFP-positive cells) and cell death rate (propidium iodide staining) were assessed by flow cytometry using a Canto 3 L flow cytometer (BD Biosciences) and BD FACSDiva Software Version 6.1.3. and analyzed with FlowJo 10.8.1 software.

Cultured VSMCs were infected at a multiplicity of infection (MOI) of 10 at 37 °C overnight in growth medium. After replacing the medium with fresh growth medium, cells were cultured for 5 additional days and then harvested for mRNA expression analysis.

In vivo transduction was performed via the jugular vein. In brief, mice were anesthetized intraperitoneally with 15 mg/ml ketamine and 0.1% xylazine. A small incision was made to expose the left jugular vein, and $10^9$ lentiviral particles were inoculated directly into the jugular vein (up to 200 µL in PBS). In vivo transduction was assessed by GFP immunostaining, and silencing efficiency by immunostaining of the lentivirus target protein.

## Histology

After sacrifice by $CO_2$ inhalation, mouse aortas were perfused with saline, fixed overnight in 10% formalin at 4 °C, and embedded in paraffin. Aortic cross sections (5 µm) were stained with hematoxylin and eosin or with modified Verhoeff elastic-Van Gieson (EVG) stain (Sigma-Aldrich). Images were acquired with a Zeiss Axio Scan.Z1 scanner and processed with NDP.view 2 V2.7.43 software. Elastic lamina breaks, defined as disruptions in elastic fibers, were

      

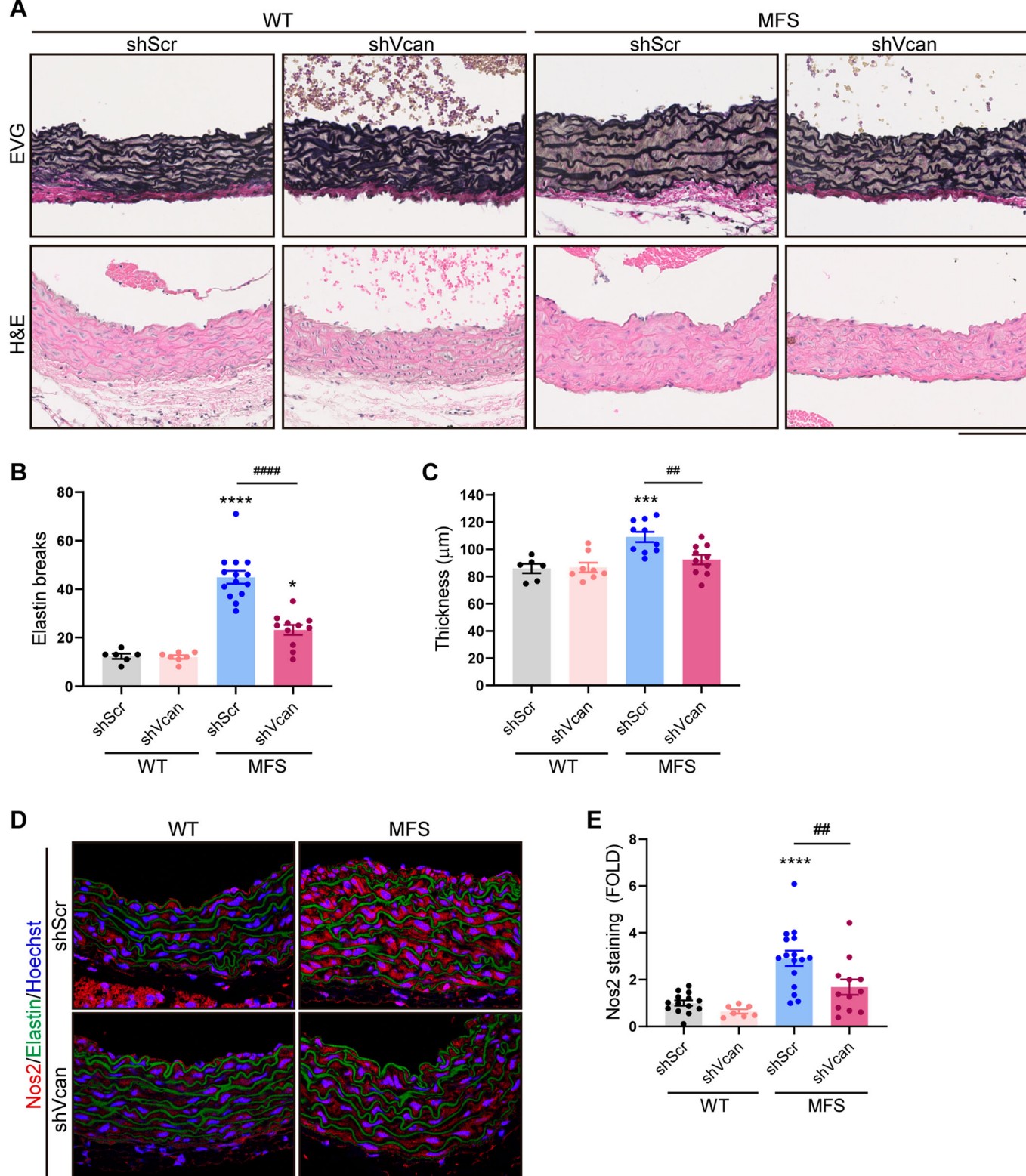

counted in the entire medial layer of three non-consecutive cross sections per mouse, and the mean number of elastin breaks per mouse was calculated. The numbers of mice per group are indicated in the figure legends.

For immunohistochemistry assays, deparaffined aortic sections were rehydrated and boiled for 3 min in citrate buffer (10 mM sodium citrate, 0.05% Tween 20, pH 6.0) to retrieve antigens. For Acan and Vcan immunohistochemistry, samples were incubated

**Figure 7. *Vcan* silencing decreases medial degeneration and Nos2 expression in MFS mice.**

(A) Representative elastic van Gieson (EVG) and hematoxylin and eosin (H&E) staining in AsAo of the indicated mice ($n = 6$–14 mice per group). Scale bar, 100 µm. (B, C) Quantification of elastin breaks (B) and medial wall thickness (C) in AsAo from the mouse cohorts shown in (A). (D) Representative images of Nos2 immunofluorescence (red), elastin autofluorescence (green), and Hoechst-stained nuclei (blue) in mouse aortic sections. Scale bar, 50 µm. The same images are also presented in the bottom line of Fig. EV3A. (E) Quantification of Nos2 immunofluorescence in AsAo from WT shScr ($n = 14$), WT sh*Vcan* ($n = 7$), MFS shScr ($n = 16$), and MFS sh*Vcan* ($n = 12$) mice. Data information: (B, C) data are shown as mean ± s.e.m. Each data point denotes an individual mouse. ***$P < 0.001$, ****$P < 0.0001$ versus shScr- transduced WT mice; ##$P < 0.01$, ####$P < 0.0001$ versus shScr-transduced MFS mice (two-way ANOVA with Tukey's post hoc test). (E) Data are shown relative to shScr-transduced WT mice as mean ± s.e.m. Each data point denotes an individual mouse. ****$P < 0.0001$ versus shScr- transduced WT mice; ##$P < 0.01$ versus shScr-transduced MFS mice (two-way ANOVA with Tukey's post hoc test). Source data are available online for this figure.

with 0.2 UI/ml chondroitinase ABC (Sigma, C3667) and 0.2 UI/ml heparinase (Sigma, H8891) in digestion buffer (20 mM Tris HCl pH8, 50 mM CaCl₂, 50 mM sodium acetate) for 30 min at 37 °C to remove GAGs. Aortic sections were then blocked for 1 h with 5% goat serum plus 2% BSA in PBS containing 0.05% Triton X-100 and incubated with the following antibodies: rabbit polyclonal anti-Acan (1/200, AB1031, Millipore), rabbit polyclonal anti-GFP (1/100, A-11122, Thermo Fisher), rabbit polyclonal anti-Vcan (1/100, AB1033, Millipore), and rabbit polyclonal anti-neo-Vcan (1/50, ab19345, Abcam), which recognizes the specific DPEAAE epitope corresponding to Adamts1-mediated cleavage of Vcan. Specificity was determined by replacing the primary antibody with the same concentration of an unrelated IgG (Santa Cruz). Color was developed with DAB (Vector Laboratories; Burlingame, CA, USA), and sections were counterstained with hematoxylin and mounted in DPX (Casa Álvarez, Madrid, Spain). Images were acquired using a Zeiss Axio Scan.Z1 scanner and NDP.view 2 V2.7.43 software.

For p-Akt and Nos2 immunofluorescence assays, deparaffined sections were rehydrated, boiled to retrieve antigens for 3 min in citrate (10 mM sodium citrate, 0.05% Tween 20, pH 6.0) or EDTA buffer (10 mM Tris base, 1 mM EDTA, 0.5% Tween 20, pH 9.0), respectively, and blocked 1 h with 5% goat serum plus 2% BSA in PBS containing 0.05% Triton X-100. Aortic sections were then incubated with rabbit polyclonal anti-Nos2 (1/200, ab15323, Abcam), rabbit monoclonal anti-p-Akt (S473) (1/50, #4060, Cell Signaling), mouse monoclonal anti-Actin α-Smooth Muscle-Cy3 antibody (1/3000, C6198, Sigma), or rat monoclonal anti-Platelet endothelial cell adhesion molecule (Cd31, Pecam-1) (1/50, DIA-310, Dianova). Specificity was determined by substituting the primary antibody with the same concentration of an unrelated IgG (Santa Cruz). Polyclonal Alexa-Fluor-647-conjugated goat anti-rabbit and polyclonal Alexa-Fluor-568-conjugated goat anti-rat (1/500, Molecular Probes) were used as secondary antibodies. Nuclei were stained with Hoechst (Sigma, B2261), and aortic sections were mounted in Citifluor AF4 mounting medium (Aname, 17973). Images were acquired with a Plan-Apochromat 40x NA 1.4 oil-immersion objective using a Zeiss LSM 700 confocal system coupled to an inverted Axio Observer Z1 microscope (ZEISS Microscopy, Jena, Germany) and Zeiss ZEN software (version 2011, 64 bits). Images ($512 \times 512$ or $1024 \times 1024$ pixels, 8 bits) were processed, analyzed, and quantified with ImageJ software (version 1.53 f).

Immunohistochemical and immunofluorescent signals were quantified after setting an intensity threshold to include only specific signals. Total intensity was corrected for the medial aortic area and normalized relative to WT or control mice, depending on the experiment.

## Immunoblot analysis

After isolation and removal of the adventitial layer, mouse aortas were frozen in liquid nitrogen. Frozen aortas were homogenized (MagNalyzer, Roche) in RIPA buffer (150 mM NaCl, 10 mM Tris HCl pH8, 1% NP-40, 0.1% SDS, and 0.5% sodium deoxycholate) supplemented with protease, phosphatase, and kinase inhibitors (100 µM benzamidin, 1 µg/ml leupeptin, 1 µg/ml pepstatin, 1 µg/ml aprotinin, 1 µM ditiotreitol, 1 mM PMSF, and 3 mM EGTA). VSMCs were washed with ice-cold PBS and lysed in RIPA buffer.

Protein extracts from aortic tissue or VSMCs were separated under reducing conditions on SDS-polyacrylamide gels and transferred to nitrocellulose membranes. Proteins were detected with the following primary antibodies: rabbit polyclonal anti-neo-Acan (specific NITEGE sequence generated by Adamts1 cleavage of Acan, PA1-1746, Thermo Fisher, 1/500), anti-neo-Vcan (specific DPEAAE neoepitope generated by Adamts1 cleavage of Vcan, ab19345, Abcam, 1/1000), rabbit monoclonal anti-p-Akt (S473) (1/1000; #4060, Cell Signaling), HRP-conjugated anti-Akt (1/1000; #8596, clone C67E7, Cell Signaling), and mouse monoclonal anti-α-tubulin (1/40,000; T6074, Sigma) and anti-Gapdh (1/20,000; ab8245, Abcam). Anti-neo-Acan antibody detects the neoepitope sequence NITEGE generated by Adamts1 cleavage (Suna et al, 2018). Bound antibodies were detected with enhanced-chemiluminescence (ECL) detection reagent (Millipore). All uncropped blots are presented in Fig. EV3.

## Real-time and quantitative PCR

Total RNA was isolated from VSMCs with TRIzol (Invitrogen). Total RNA (2 µg) was reverse-transcribed at 37 °C in a 20-µl reaction mix containing 200 U Moloney murine leukemia virus (MMLV) reverse transcriptase (Promega), 100 ng random primers, and 40U RNase inhibitor (Invitrogen). Real-time quantitative RT-PCR was performed with PCR primers for *Acan* (CCTGCTACTT CATCGACCCC, AGATGCTGTTGACTCGAACCT), *Vcan* (CTGATAGCAGATTTGATGCCTACTGC, GTGGTTCTTTGGA TAAACTGGGTGATG), and *Gapdh* (TGACGTGCCGCCTGGA GAAA, AGTGTAGCCCAAGATGCCCTTCA). qPCR reactions were performed in triplicate with SYBR master mix (Promega). Probe specificity was assessed in a post-amplification melting-curve analysis, with only one melting-temperature (Tm) peak being produced for each reaction. The amount of target mRNA was estimated with the $2^{-\Delta\Delta CT}$ relative quantification method and normalized to the expression of *Gapdh*. qPCR data were analyzed with Bio-Rad CFX Manager 3.1 software, with further analysis in Microsoft Excel.

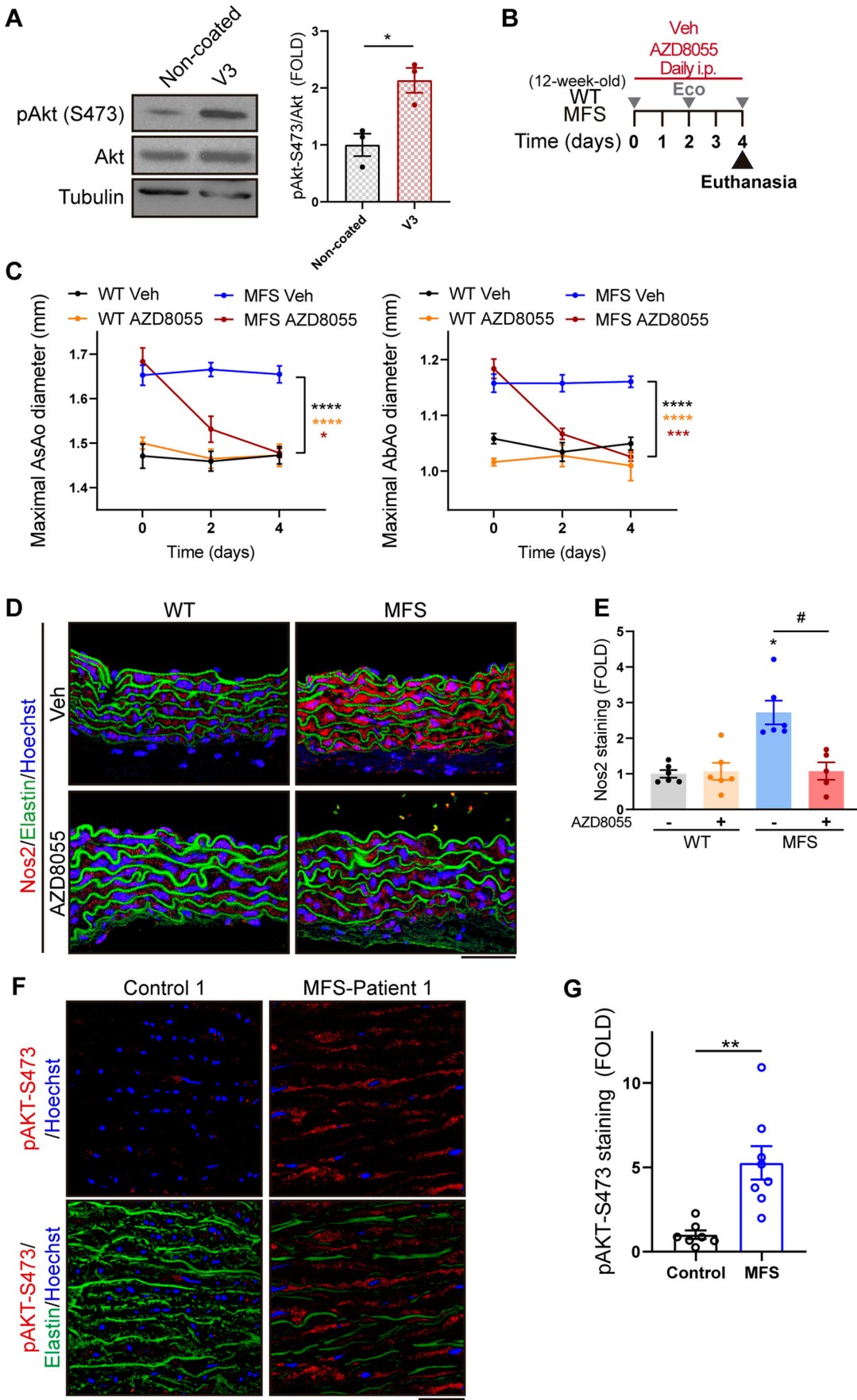

**Figure 8. VCAN triggers aortic dilation via Akt activation and Nos2 induction in MFS mice.**

(A) Representative immunoblot analysis (left panel) and quantification of the immunoblot signal (right panel) of p-Akt-S473, total Akt, and tubulin in protein extracts from VSMCs grown in serum-starved conditions overnight and subsequently seeded for 1 h on plates precoated with 20 µg/ml VCAN V3 or treated with PBS as a control (n = 3 independent experiments). (B) Experimental design: 12-week-old WT and MFS mice were treated on 4 consecutive days with 20 mg/kg/day AZD8055. Longitudinal ultrasound (Eco) was performed at the indicated time points, and mice were euthanized after 4 days of treatment. (C) Maximal AsAo and AbAo diameters of WT Veh (n = 6), WT AZD8055 (n = 5), MFS Veh (n = 6), and MFS AZD8055 (n = 6) mice at the indicated time points. (D) Representative images of Nos2 immunofluorescence (red), elastin autofluorescence (green), and Hoechst-stained nuclei (blue) in mouse aortic sections. Scale bar, 50 µm. The same images are also presented in the bottom line of Fig. EV4. (E) Quantification of Nos2 immunofluorescence in AsAo from WT Veh (n = 6), WT AZD8055 (n = 6), MFS Veh (n = 6), and MFS AZD8055 (n = 5) mice. (F) Representative images of pAKT-S473 immunofluorescence (red), elastin autofluorescence (green), and Hoechst-stained nuclei (blue) in the medial layer of human aortic tissue from control donors and MFS patients. Scale bar, 50 µm. The same images are also presented in the first line of Fig. EV5. (G) Quantification of pAKT-S473 immunofluorescence in aortas from control donors (n = 7) and MFS patients (n = 8). Data information: (A) data are shown relative to non-coated plates (control) as mean ± s.e.m. Each data point denotes an independent experiment. *P < 0.05 (Student t test). (C) Data are shown as mean ± s.e.m. *P < 0.05, ***P < 0.001, ****P < 0.0001 versus MFS Veh mice (repeated-measurements two-way ANOVA with Tukey's post hoc test). (E) Data are shown relative to Veh-treated WT mice as mean ± s.e.m. Each data point denotes an individual mouse. *p < 0.05 versus Veh-treated WT mice; #P < 0.05 versus Veh-treated MFS mice (Kruskal–Wallis test with Dunn's multiple comparison test). (G) Data are shown relative to control donors as mean ± s.e.m. Each data point denotes an individual. **P < 0.01 (unpaired t test with Welch's correction). Source data are available online for this figure.

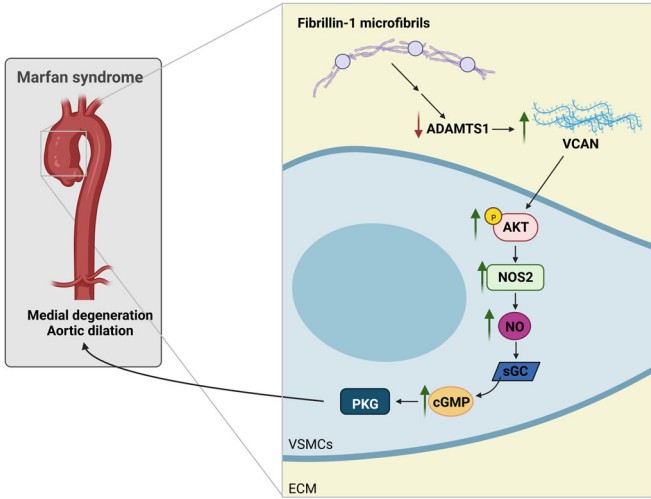

**Figure 9. Model depicting the contribution of VCAN and AKT to the aortic phenotype in MFS.**

Mutations in the *FBN1* gene lead to ADAMTS1 deficiency, resulting in VCAN accumulation, which activates the AKT signaling pathway. The resulting induction of NOS2 leads to the production of supraphysiological concentrations of NO and overactivation of the canonical NO signaling pathway, which ultimately leads to aortic dilation and medial degeneration.

## Plasma Vcan levels

Vcan levels were quantified in mouse plasma using a mouse Vcan ELISA kit (ab279409, Abcam; Cambridge, UK). To obtain mouse plasma, blood was extracted by the submandibular bleeding method, collected in heparin-containing tubes, and centrifuged for 10 min at 2000×g.

## Human samples

The study complied with all relevant ethical regulations and was approved by the Clinical Research Ethics Committee of Cantabria (ref. 27/2013) and the Ethics Committee of Instituto de Salud Carlos III (CEI PI91_2018-v2-Enmienda_2019) conformed to the principles set out in the WMA Declaration of Helsinki

and the Department of Health and Human Services Belmont Report. Ascending aorta samples used as controls were obtained anonymously from multiorgan transplant donors. During preparation of the heart for transplantation, excess AsAo tissue was trimmed and harvested for the study. Samples and data from MFS patients included in this study were obtained during elective or emergency surgery for aortic root replacement and provided by the Hospital Universitari Vall d'Hebron Biobank (National Registry of Biobanks B.000018, PT20/00107). Control and MFS samples were obtained from male and female individuals aged between 27 and 79 years old. Tissues were immediately fixed, kept at room temperature for 48 h, and embedded in paraffin. Informed consent was obtained from all human participants or their families. Patient clinical data were retrieved while maintaining anonymity.

## Statistical analysis

GraphPad Prism 8.4.3 was used for statistical analysis. Outliers were identified and excluded by the ROUT method. Data normality was assessed by the Shapiro–Wilk test, and appropriate tests were chosen according to data distribution. Differences were analyzed by the two-tailed Student or the Welch t test (depending on the standard deviation of the tested conditions); two-way or repeated-measurements two-way analysis of variance (ANOVA) with the Tukey post hoc test (experiments with ≥3 groups); or the Kruskal–Wallis test with Dunn's multiple comparison test, as appropriate. Statistical significance was assigned at *P < 0.05, **P < 0.01, ***P < 0.001, and ****P < 0.0001. Unless otherwise stated in the figure legends, asterisks (*) indicate comparison with the control or WT phenotype, whereas hash signs (#) indicate comparison with the MFS phenotype.

The numbers of animals used are indicated in the corresponding figure legends. Sample size was chosen empirically based on our previous experience in the calculation of experimental variability. All experiments were performed with at least three biological replicates. No randomization was performed to allocate animals into experimental groups, and investigators were not blinded to group allocation during in vivo experiments. Experimental groups were balanced in terms of animal age and randomly assigned to the experimental treatments. Animals were genotyped before experiments, caged together, and treated in the same way.

 

## The paper explained

### Problem

Marfan syndrome (MFS) is a syndromic disease caused by mutations in the gene encoding fibrillin-1. Thoracic aortic aneurysms and dissections (TAAD) are the leading cause of morbidity and mortality in individuals with MFS. Both mice and patients with MFS exhibit elevated NOS2 and decreased ADAMTS1 protein levels in the aorta. However, the underlying mechanisms triggering aortic induction of NOS2 in MFS remained elusive.

### Results

Versican (Vcan), an ADAMTS1 substrate, accumulates in the aortas of both MFS mice and patients, and plays a central role in mediating MFS aortopathy. Indeed, knockdown of *Vcan* regresses aortic dilation and medial degeneration by reducing Nos2 overexpression in MFS mice. Our findings demonstrate that Vcan induces Akt activation. Pharmacological inhibition of the Akt signaling effectively reverses aortic dilatation and restores Nos2 expression in MFS mice. Furthermore, patients with MFS show increased pAKT-S473 levels, indicative of elevated AKT activation.

### Impact

Our findings identify the mTOR-Akt pathway as a promising therapeutic target for addressing MFS-related aortopathy. This discovery underscores the need to assess the potential of AKT inhibitors as a therapeutic option for patients with MFS.

## Data availability

This manuscript does not have data that needs to be deposited in a public database.

## Peer review information

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

## Acknowledgements

We thank S. Bartlett for English language editing; V Labrador for advice on confocal imaging and immunofluorescent experiments; the CNIC Facilities of histology, microscopy, and advanced imaging; and AI Torralbo for excellent technical support and advice. The CNIC is supported by the Instituto de Salud Carlos III (ISCIII), the Ministerio de Ciencia e Innovación (MCIN) and the Pro CNIC Foundation), the CBMSO is supported by Consejo Superior de Investigaciones Científicas and Universidad Autónoma de Madrid. CBMSO and CNIC are Severo Ochoa Centers of Excellence (grants CEX2021-001154-S and CEX2020-001041-S, respectively) funded by MICIN/AEI/10.13039/501100011033. The project leading to these results has received funding from "La Caixa" Banking Foundation under project codes HR18-00068 (to MRC and JMR); Spanish Ministerio de Ciencia e Innovación grant RTI2018-099246-B-I00 (MICIU/AEI/FEDER, UE) to JMR, and grants PID2020-115217RB-I00 and PID2021-122388OB-I00 to MRC and JMR, respectively, funded by MCIN/AEI/10.13039/501100011033; Instituto de Salud Carlos III (CIBER-CV CB16/11/00264 and CB16/11/00479; and grants PI17/00381 to GT-T and PI21/00084 (co-funded by Fondo Europeo de Desarrollo Regional (FEDER)) to JFN); Fundacio La Marato TV3 (20151330 to JMR); Instituto de Investigación Sanitaria Marqués de Valdecilla (IDIVAL) (INNVAL 21/24) to JFN; The Marfan Foundation USA Faculty grant 2017 MRF/1701 (to JMR); Fundación MERCK-Fundación Española de Enfermedades Raras 2022 and V-Ayudas "Muévete por los que no pueden 2021" (to JO); and Spanish Ministerio de Ciencia e Innovación contracts FPI (BES-2016-077649) to MJR-R; Sara Borrell (CD18/00028) and Juan de la Cierva (IJC2020-044581-I) to MT; Ramón y Cajal (RYC2021-033343-I) to JO; and FPU (20/04814) to IA-R.

## Author contributions

**Maria Jesus Ruiz-Rodriguez**: Conceptualization; Formal analysis; Validation; Investigation; Visualization; Methodology; Writing—original draft; Writing—review and editing. **Jorge Oller**: Conceptualization; Formal analysis; Validation; Investigation; Methodology. **Sara Martínez-Martínez**: Conceptualization; Formal analysis; Validation; Investigation; Methodology; Writing—review and editing. **Iván Alarcón-Ruiz**: Validation; Investigation. **Marta Toral**: Validation; Investigation. **Yilin Sun**: Validation; Investigation. **Angel Colmenar**: Formal analysis; Validation; Investigation; Methodology. **María José Méndez-Olivares**: Formal analysis; Validation; Investigation; Methodology. **Dolores Lopez-Maderuelo**: Validation; Investigation; Methodology. **Christine B Kern**: Conceptualization; Resources. **J Francisco Nistal**: Conceptualization; Resources. **Arturo Evangelista**: Conceptualization; Resources. **Gisela Teixido-Tura**: Conceptualization; Resources. **Miguel R Campanero**: Conceptualization; Resources; Formal analysis; Supervision; Funding acquisition; Visualization; Writing—original draft; Project administration; Writing—review and editing. **Juan Miguel Redondo**: Conceptualization; Resources; Formal analysis; Supervision; Funding acquisition; Visualization; Writing—original draft; Project administration; Writing—review and editing.

## Disclosure and competing interests statement

The authors declare no competing interests.

# Expanded View Figures

**Figure EV1. Accumulation of Vcan in aortas of MFS mice at 12 weeks but not at 4 weeks of age.**                                              ▶

(A) Representative images of Vcan immunofluorescence (red), elastin autofluorescence (green), and Hoechst-stained nuclei (blue) in aortic sections from 4- or 12-week-old WT and MFS mice. Scale bar, 50 μm. (B) Quantification of Vcan immunofluorescence in mouse AsAo (4-week-old: WT, $n = 4$; MFS, $n = 3$. 12-week-old: WT, $n = 4$; MFS, $n = 3$). Data information: (B) data are shown relative to 12-week-old WT mice as mean ± s.e.m. Each data point denotes an individual mouse. *$P < 0.05$ (two-way ANOVA with Tukey's post hoc test). Source data are available online for this figure.

**A**

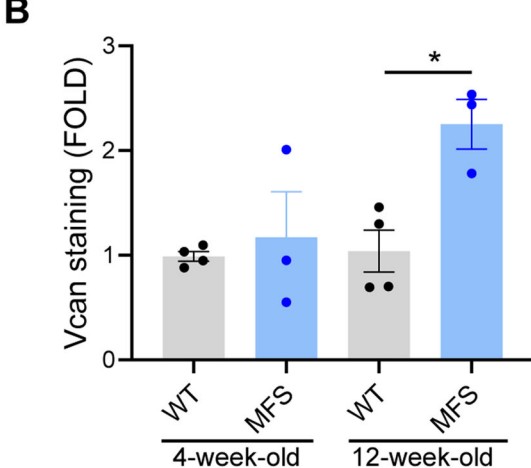

**B**

Figure EV2.

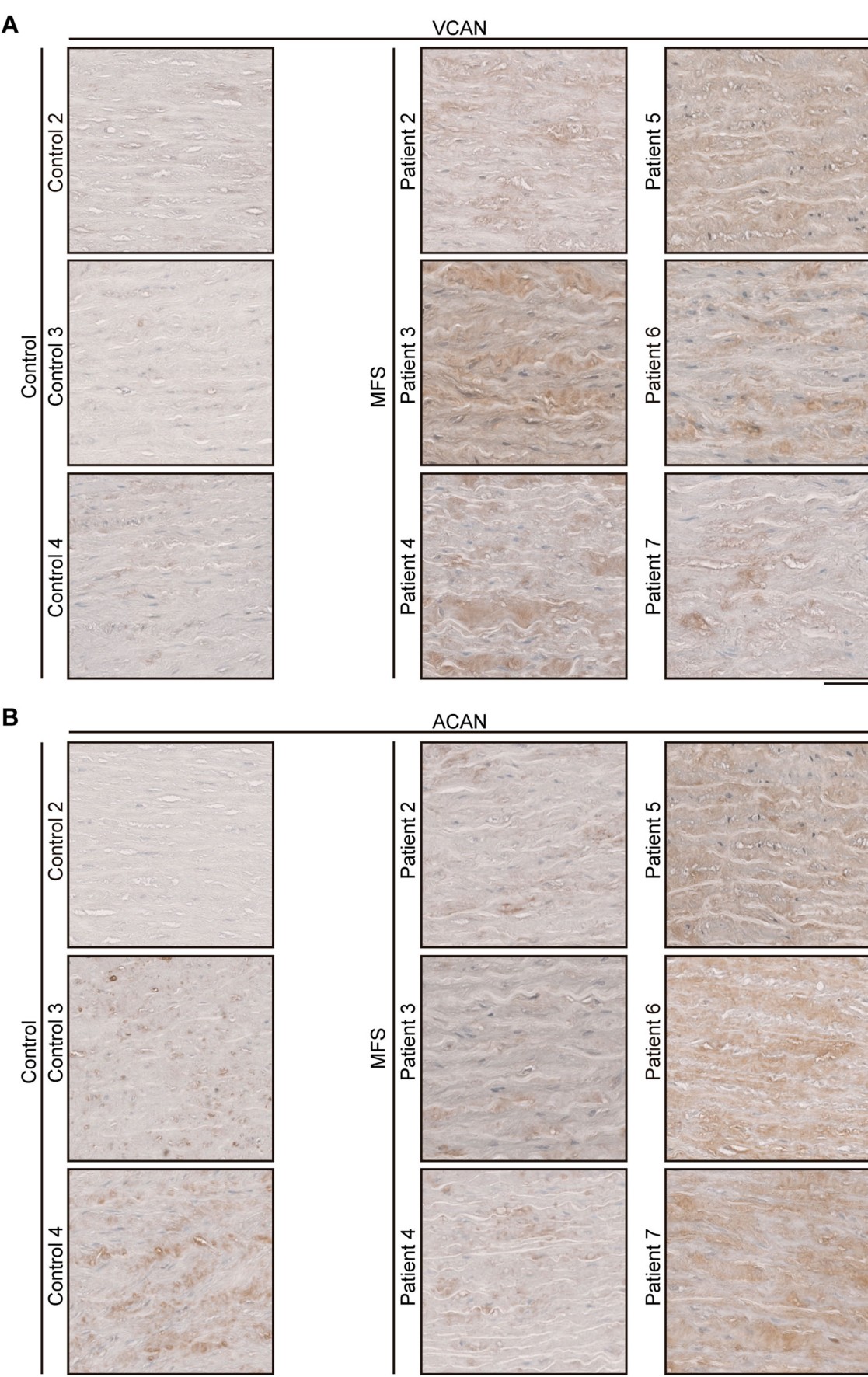

◄    **Figure EV2.   VCAN and ACAN protein expression in aortas of MFS patients.**

(**A, B**) Representative images of (**A**) VCAN and (**B**) ACAN in the medial layer of aortic sections from 3 control donors and 6 MFS patients. Scale bar, 50 μm. Source data are available online for this figure.

     *EMBO Molecular Medicine* Volume 16 | January 2024 | 132 – 157     **EV4**

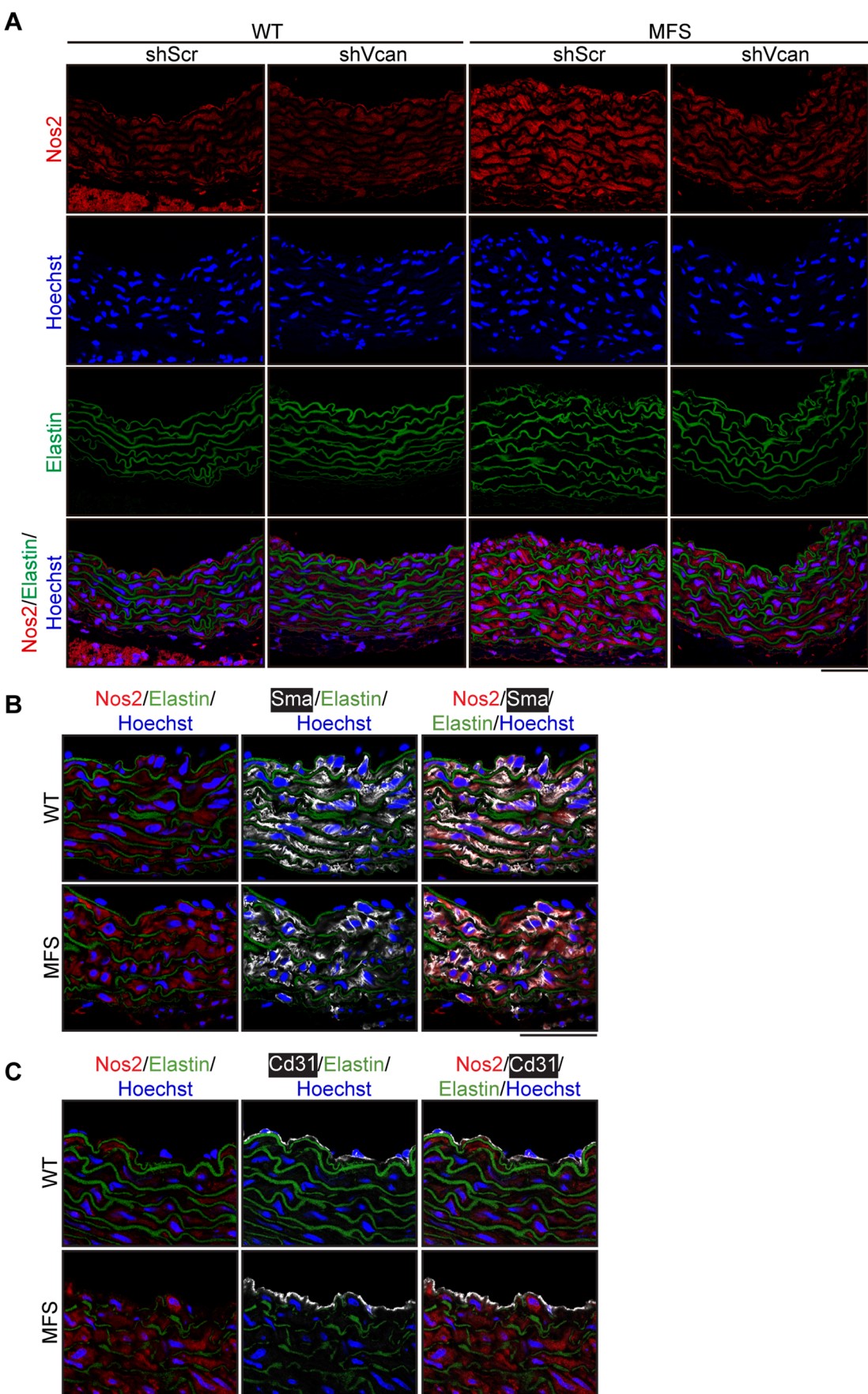

**Figure EV3.   *Vcan* silencing reduces Nos2 expression in aortas from MFS mice.**

(A) Representative images of Nos2 immunofluorescence (red), Hoechst-stained nuclei (blue), and elastin autofluorescence (green) in aortic sections from 16-week-old MFS and WT mice. Individual channels are shown followed by a composite image. Scale bar, 50 μm. The 4 merged images are identical to those shown in Fig. 7D. (B, C) Representative images of (B) Sma or (C) Cd31 (pale gray), Nos2 immunofluorescence (red), elastin autofluorescence (green), and Hoechst-stained nuclei (blue) in aortic sections from 12-week-old WT and MFS mice. Scale bar, 50 μm. Source data are available online for this figure.

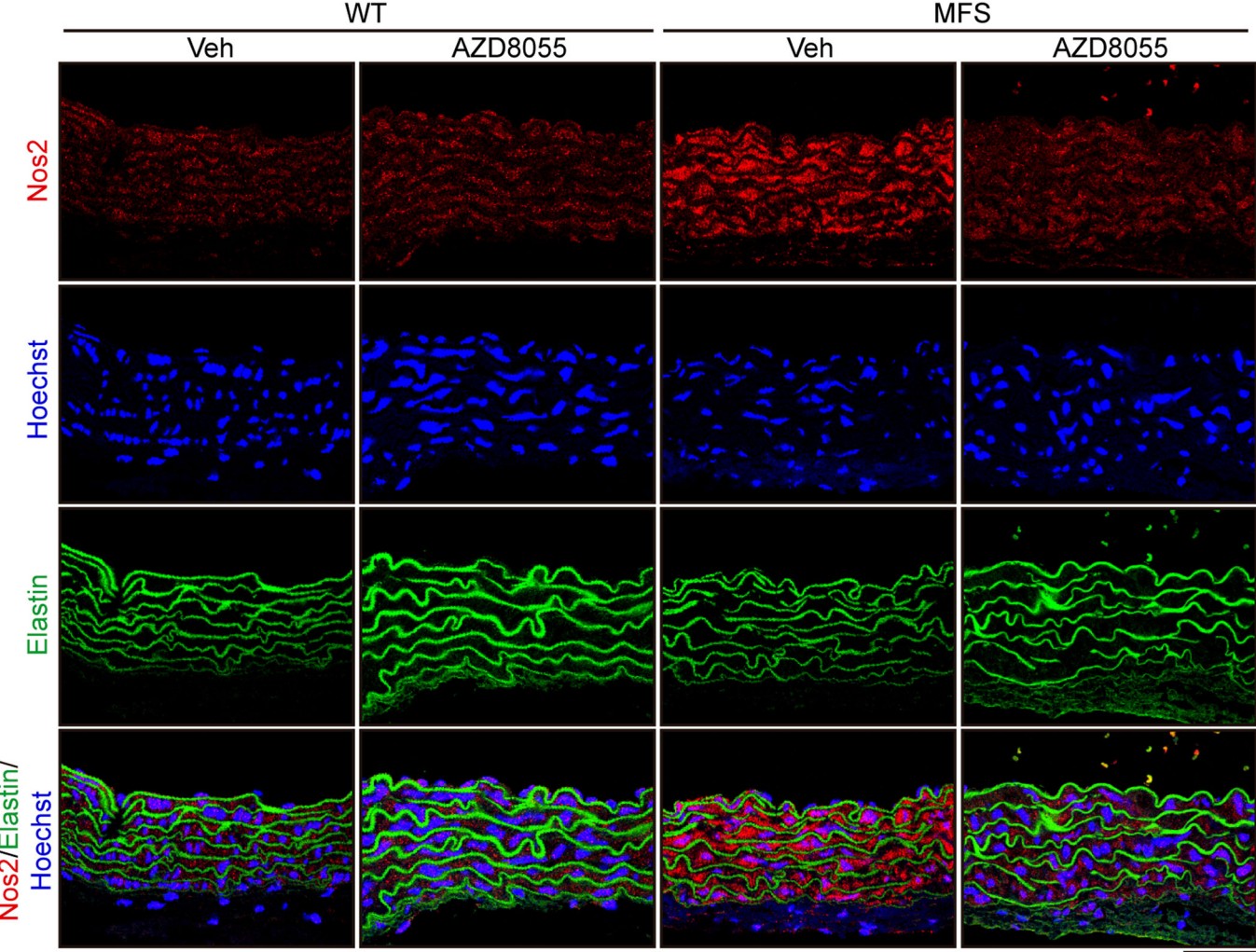

**Figure EV4. Pharmacological inhibition of Akt signaling decreases aortic Nos2 expression in MFS mice.**

Representative images of Nos2 immunofluorescence (red), Hoechst-stained nuclei (blue), and elastin autofluorescence (green), in aortic sections from WT and MFS mice treated as indicated. Individual channels are shown followed by a composite image. Scale bar, 50 μm. The 4 merged images are identical to those shown in Fig. 8D. Source data are available online for this figure.

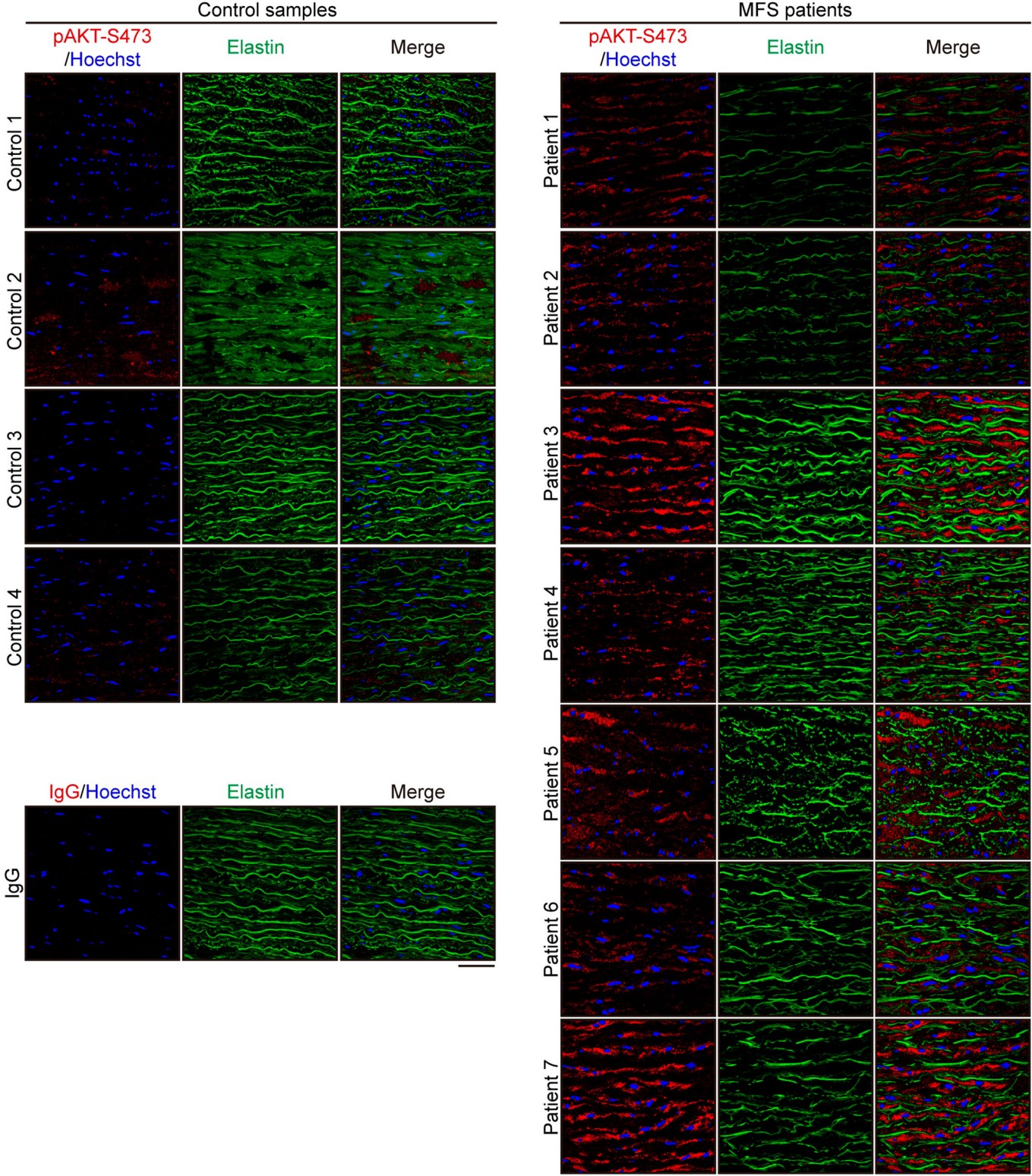

**Figure EV5. AKT is activated in the aortas of MFS patients.**

Representative images of pAKT-S473 immunofluorescence (red), Hoechst-stained nuclei (blue), and elastin autofluorescence (green) in the medial layer of aortic sections from 4 control donors and 7 MFS patients. A representative image of the staining with a control IgG (red) is also shown. Scale bar, 50 μm. The pAKT-S473/Hoechst and merge images corresponding to Control 1 and Patient 1 are identical to the pAKT-S473/Hoechst and pAKT-S473/Elastin/Hoechst images shown in Fig. 8F. Source data are available online for this figure.

