## [Peer Review File · EMBO Molecular Medicine]

Versican accumulation drives Nos2 induction and aortic disease in Marfan syndrome via Akt activation

Maria Jesus Ruiz-Rodriguez, Jorge Oller, Sara Martínez-Martínez, Iván Alarcón-Ruiz, Marta Toral, Yilin Sun, Angel Colmenar, María José Méndez-Olivares, Dolores Lopez-Maderuelo, Christine Kern, J. Francisco Nistal, Arturo Evangelista, Gisela Teixido-Tura, Miguel Campanero, and Juan Miguel Redondo

DOI: [10.15252/emmm.202317799](https://doi.org/10.15252/emmm.202317799)

Corresponding authors: [Juan Miguel Redondo \(jmredondo@cnic.es\)](mailto:jmredondo@cnic.es) , [Miguel Campanero \(mcampanero@cbm.csic.es\)](mailto:mcampanero@cbm.csic.es)

Review Timeline:

Submission Date:	3rd Apr 23
Editorial Decision:	3rd Jul 23
Revision Received:	11th Oct 23
Editorial Decision:	1st Nov 23
Revision Received:	2nd Nov 23
Accepted:	10th Nov 23

Editor: *Kelly Anderson*

Transaction Report:

3rd Jul 2023

Dear Dr. Redondo,

Thank you for the submission of your manuscript to EMBO Molecular Medicine. We have now received feedback from the three reviewers who agreed to evaluate your manuscript. As you will see from the reports below, the referees acknowledge the interest of the study and are overall supporting publication of your work pending appropriate revisions. It would be good to discuss your plan to address the referee concerns and I am available to do so in the coming weeks by email or zoom.

Addressing the reviewers' concerns in full will be necessary for further considering the manuscript in our journal, and acceptance of the manuscript will entail a second round of review. EMBO Molecular Medicine encourages a single round of revision only and therefore, acceptance or rejection of the manuscript will depend on the completeness of your responses included in the next, final version of the manuscript. For this reason, and to save you from any frustrations in the end, I would strongly advise against returning an incomplete revision.

I look forward to seeing a revised form of your manuscript as soon as possible. Use this link to login to the manuscript system and submit your revision: <https://embomolmed.msubmit.net/cgi-bin/main.plex>

Yours sincerely,

Kelly

Kelly M Anderson, PhD
Scientific Editor
EMBO Molecular Medicine

We require:

2) Individual production quality figure files as .eps, .tif, .jpg (one file per figure). For guidance, download the 'Figure Guide PDF': (<https://www.embopress.org/page/journal/17574684/authorguide#figureformat>).

3) A .docx formatted letter INCLUDING the reviewers' reports and your detailed point-by-point responses to their comments. As part of the EMBO Press transparent editorial process, the point-by-point response is part of the Review Process File (RPF), which will be published alongside your paper.

4) A complete author checklist, which you can download from our author guidelines (<https://www.embopress.org/page/journal/17574684/authorguide#submissionofrevisions>). Please insert information in the checklist that is also reflected in the manuscript. The completed author checklist will also be part of the RPF.

6) It is mandatory to include a 'Data Availability' section after the Materials and Methods. Before submitting your revision, primary datasets produced in this study need to be deposited in an appropriate public database, and the accession numbers and database listed under 'Data Availability'. Please remember to provide a reviewer password if the datasets are not yet public (see <https://www.embopress.org/page/journal/17574684/authorguide#dataavailability>).

7) For data quantification: please specify the name of the statistical test used to generate error bars and P values, the number (n) of independent experiments (specify technical or biological replicates) underlying each data point and the test used to calculate p-values in each figure legend. The figure legends should contain a basic description of n, P and the test applied.

Graphs must include a description of the bars and the error bars (s.d., s.e.m.). See also 'Figure Legend' guidelines: <https://www.embopress.org/page/journal/17574684/authorguide#figureformat>

.

13) Author contributions: You will be asked to provide CRediT (Contributor Role Taxonomy) terms in the submission system. These replace a narrative author contribution section in the manuscript.

14) A Conflict of Interest statement should be provided in the main text.

EMBO Molecular Medicine has a "scooping protection" policy, whereby similar findings that are published by others during

review or revision are not a criterion for rejection. Should you decide to submit a revised version, I do ask that you get in touch after three months if you have not completed it, to update us on the status.

Please note: When submitting your revision you will be prompted to enter your funding and payment information. This will allow Wiley to send you a quote for the article processing charge (APC) in case of acceptance. This quote takes into account any reduction or fee waivers that you may be eligible for. Authors do not need to pay any fees before their manuscript is accepted and transferred to the publisher.

EMBO Press participates in many Publish and Read agreements that allow authors to publish Open Access with reduced/no publication charges. Check your eligibility: <https://authorservices.wiley.com/author-resources/Journal-Authors/open-access/affiliation-policies-payments/index.html>

**** Reviewer's comments ****

Referee #1 (Remarks for Author):

The manuscript of Ruiz-Rodriguez et al is entitled « Versican accumulation drives Nos2 induction and aortic disease in Marfan syndrome via Akt activation. » The authors recently involved ADAMTS1 downregulation and NOS2 induction in aortopathy developed in Marfan syndrome But the link between ADAMTS1 and NOS1 remained unknown. The authors hypothesize that the accumulation of uncleaved ADAMTS1 substrates may be one of the links. They have shown that Versican accumulates in the aortas of Marfan patients and Fbn1 C1041G/+ mice and demonstrate that this proteoglycan accumulation is correlated with the presence of Nos2 in aortas and Akt activation.

Several questions need to be addressed before the authors can support their conclusion :

Figure 4 they estimated that Acan is not involved because after euthanasia at 28 days, they do not observe any differences. The fig4C is not convincing. For the WT +shScr, the aorta is larger than in MFs sample. The immunostaining seems to be intracellular and not extracellular. The fig4D is more convincing except for the WT+shAcan. This picture seems to be at a higher magnification. In Fig 4A the Acan silencing is obvious in cells but not in histology (fig4C). Why is there any difference between AbAo and ASAo in histology (Fig4C)?

During the progression of aortic dilatation, at which point the cells start to increase the production of Vcan and Acan ? Do the level of this PG expression be higher and induced by the genetic mutation at birth? At one month old ? The human sample can not give this information.

Figure 6 how could they explain the absence of the shVcan efficiency in WT mice ?

Fig 7 the authors have shown the decrease of Nos2 after the addition of shVcan at protein level. what about the gene expression level of Vcan?

Is it possible to provide the distinct stainings (elastin, Nos2 and Hoechst) separately?

Which type of cells synthesized the NOS2 ? Where does the NOS2 come from ? Endothelial cell or Smooth muscle cells ?

How do the authors explain the potential link of Nos 2 in intracellular with the Vcan (extracellular) ?

Fig 8 there is a problem with the image D on the right panel.

In A, is it possible to quantify the difference of pAkt between non coated and coated with Versican ?

In their conclusion, in Fig 9, they drew a schema with ADAMTS1. Are they sure that it is not another ADAMTS responsible of the phenotype ? What are the level of ADAMTS 4 and 5, other versicanases in the aortas of Fbn1 C1041G/+?

In the introduction, line 98, put C1039G/+ as the exponent of Fbn1 and change C1039 to C1041 (the correct number in mouse).

Referee #2 (Comments on Novelty/Model System for Author):

The mouse model used in this manuscript is widely acceptable and well-characterized that greatly representative of the progression of the aortopathy of this rare disease. In addition, authors use isolated vascular aortic smooth muscle cells to evaluate some mechanistic process.

Referee #2 (Remarks for Author):

The manuscript presented by Redondo and coworkers introduce versican (but not aggrecan) as a new driver of the aortopathy progression in Marfan syndrome taking advantage of respective subproducts generated by ADAMTS1, which is known to be downregulated in Marfan aorta. They nicely utilize new mouse models (Fbn1C1041G/+Vcanhdf/+) and (Fbn1C1041G/+Acancmd/+) to evaluate their respective impact in the aneurysm formation. Likewise, they specific target to aorta lentiviral tools against the expression of both proteoglycans.

The manuscript is very well written and nice to read. Introduction is clear and rationale of experiments and tools used are also clear. I would suggest that in the panel where they represent the experimental design using lentiviral injections (Figs 4B and 8B) they clearly indicate the age at which animals are injected (12 weeks age) regardless it is already mentioned in the figure legend and/or in the text. Many times, readers directly look at the figures to follow results ignoring the reading of figure legends (usually

long or very long ones).

Results are very well described and clear. Everything fits very well, but there are only two things that somewhat worry me about.

1. The cellular localization of Vcan (fig. 1B) and the nuclear-like staining for the presence of both proteoglycans (Figs 4 and 6), which is not the case in human aortae (EV1). I would expect to visualize the HRP staining at the extracellular space and not intracellularly, and at lesser extent into the nucleus.
2. The unexpected regression of the aortopathy (both thoracic and abdominal) following lentiviral injections (already evident at the 3rd day after injection and being fully restored after the week/7 days). It would be interesting to have transmission electron microscopy images of aortae before and after lentiviral injections to know in detail to what extent the aortic architecture is normalized. In this respect, ultrasound images do not resolve enough.
3. It is rather surprising that the treatment with AZD8055 (inhibitor of Akt signaling) for 2 and 4 days is able to revert the aortopathy, when usually most of pharmacological treatments takes longer to revert aortic injuries in the same MFS mouse model (and in other too mgR/mgR for example). It would be interesting to read a comment in the Discussion in this respect. Finally, looking the nice EV2 images, it seems rather clear that there is no apparent association between the expression of pAKT-S473 and the level of disarray of elastic laminae. I think that this deserves some commentary in the Discussion.

Referee #3 (Comments on Novelty/Model System for Author):

A well performed study with different models showing the same outcome, so it feels robust.

Referee #3 (Remarks for Author):

Versican accumulation drives Nos2 induction and aortic disease in Marfan syndrome via Akt activation.

A clear study to find the missing link between the ADAMTS1 and iNOS dysregulation, which the authors published previously. Signs from the literature were already present, yet the dots are now connected by VCAN. I do not have much to add to the data itself, however, a number of interesting references are lacking, where hidden clues on dysregulated VCAN or Akt signaling in Marfan were already provided, which should be mentioned in the discussion.

Arterioscler Thromb Vasc Biol. 2023 Jul;43(7):1134-1153. doi: 10.1161/ATVBAHA.122.318448.

Arterioscler Thromb Vasc Biol. 2019 Sep;39(9):1859-1873. doi: 10.1161/ATVBAHA.118.312175.

Matrix Biol. 2022 Jun;110:106-128. doi: 10.1016/j.matbio.2022.05.002.

Clin Epigenetics. 2021 Dec 11;13(1):217. doi: 10.1186/s13148-021-01204-4.

How should we interpret the data with regard to the eNOS/Akt study in endothelial cells in MFS mice? Br J Pharmacol. 2007 Apr;150(8):1075-83. doi: 10.1038/sj.bjp.0707181. Could it be the loss of eNOS and Akt signaling there that triggers the SMC to attempt to rescue the lack of NO? In other words, would there be crosstalk as to how the pathological process starts? Is eNOS/pAkt recovered in endothelial cells upon VCAN haploinsufficiency? Can the authors see this in their IHC or IF stainings?

Can VCAN (or ACAN) be found in plasma or serum of the mice and is there a correlation between these levels and aortic dilatation? That would strengthen their discussion on this topic.

***** Reviewer's comments *****

Referee #1 (Remarks for Author):

The manuscript of Ruiz-Rodriguez et al is entitled « Versican accumulation drives Nos2 induction and aortic disease in Marfan syndrome via Akt activation. » The authors recently involved ADAMTS1 downregulation and NOS2 induction in aortopathy developed in Marfan syndrome But the link between ADAMTS1 and NOS1 remained unknown. The authors hypothesize that the accumulation of uncleaved ADAMTS1 substrates may be one of the links. They have shown that Versican accumulates in the aortas of Marfan patients and Fbn1 C1041G/+ mice and demonstrate that this proteoglycan accumulation is correlated with the presence of Nos2 in aortas and Akt activation.

Several questions need to be addressed before the authors can support their conclusion :

Figure 4 they estimated that Acan is not involved because after euthanasia at 28 days, they do not observe any differences. The fig4C is not convincing. For the WT +shScr, the aorta is larger than in MFS sample. The immunostaining seems to be intracellular and not extracellular. The fig4D is more convincing except for the WT+shAcan. This picture seems to be at a higher magnification. In Fig 4A the Acan silencing is obvious in cells but not in histology (fig4C). Why is there any difference between AbAo and ASAo in histology (Fig4C)?

We appreciate the constructive feedback from the Reviewer, and we have carefully considered the important issues raised to enhance the clarity and accuracy of our manuscript.

We regret any misunderstanding that may have arisen regarding Figure 4C. It is important to clarify that Figure 4C actually shows GFP immunostaining, which was performed to confirm aortic transduction following lentiviral administration. Acan immunostaining is presented in Figure 4D. For clarity, we have now emphasized "GFP" in bold green within Figure 4C to avoid any potential misunderstanding.

We have carefully reviewed all images in Figure 4D and confirmed that they are at the same magnification. Any perceived differences in magnification with the WT shAcan sample are likely due to the non-perpendicular orientation of the aortic wall. In the revised **Figure 4D**, we have outlined the boundaries between the media and the adventitia with dotted lines, providing greater consistency in the size of the medial layer of WT aortas and highlighting the expected increase in thickness in MFS mouse aortas. Indeed, we have included a black dotted line between the adventitial and medial layers in all mouse immunohistochemical stains within the manuscript to facilitate the differentiation between these two layers.

Regarding the variations between ASaO and AbAo in Figure 4C, these images illustrate the inherent structural histological distinctions between these aortic regions.

During the progression of aortic dilatation, at which point the cells start to increase the production of Vcan and Acan ? Do the level of this PG expression be higher and induced by the genetic mutation at birth? At one month old ? The human sample can not give this information.

The Reviewer is right regarding the limited informativeness of human samples in understanding the dynamics of Vcan accumulation in MFS. Given our data showing the early accumulation of Vcan, but not Acan, in the aortas of MFS mice at 12 weeks of age, we have conducted Vcan staining in aortic cross sections from both 4- and 12-week-old WT and MFS mice. We have included new data in the revised

version of the manuscript (Results, page 6, paragraph 1) to demonstrate that Vcan accumulation is higher in the aortas of 12-week-old mice and is similar in WT and MFS aortas at 4 weeks of age (**new Figure EV1**). Our findings suggest that Vcan accumulation in the aortic wall of MFS mice increases sometime between 4 and 12 weeks of age.

Figure 6 how could they explain the absence of the shVcan efficiency in WT mice ?

The Reviewer raises an interesting issue (which we had also considered) for which we currently lack an experimentally established answer. However, we have proposed a speculative explanation that centers on the turnover of Vcan. It is plausible that Vcan turnover differs under physiological conditions compared to the context of MFS aortic disease. If a much faster turnover occurs under pathological conditions in the aortas of MFS mice than in those of WT mice, it could account for why Vcan silencing selectively decreases Vcan accumulation in MFS aortas. We now include a commentary on this issue on page 8, paragraph 1.

Fig 7 the authors have shown the decrease of Nos2 after the addition of shVcan at protein level. what about the gene expression level of Vcan?

In our experience, it is frequently observed that protein levels do not correlate with mRNA levels, as we have encountered when attempting to validate protein changes identified in transcriptomic analyses. Therefore, whenever feasible, we prioritize the analysis of protein levels, as in this study where the efficacy of Vcan silencing was validated through immunohistochemistry (Figures 6D and 6E). We believe that protein expression is more biologically decisive, and its validation helps circumvent issues derived from post-transcriptional mRNA regulation.

Is it possible to provide the distinct stainings (elastin, Nos2 and Hoechst) separately?

Following the Reviewer suggestion, we have created separated images for elastin, Hoechst, and Nos2 or pAKT-S473, which are presented in the **new Figure EV3A** and the **new Figure EV4**. We greatly appreciate this suggestion, as displaying each staining individually markedly enhances the clarity and distinction of each specific signal.

Which type of cells synthesized the NOS2 ? Where does the NOS2 come from ? Endothelial cell or Smooth muscle cells ?

This is indeed a relevant question, and we concur with the Reviewer regarding the significance of identifying the cellular source of Nos2. In the revised version, we have included immunofluorescence data on aortic cross sections, which demonstrate that elevated Nos2 levels in MFS aortas are predominantly detected in smooth muscle cells (Sma)-positive cells (**new Figure EV3B**). However, we do not exclude the possibility that some endothelial cells may also express Nos2 (**new Figure EV3C**). These additional findings are described in the Results section (page 8, last paragraph) and align with previous results from our research group, showing increased NOS2 expression in vascular smooth muscle cells in patients with MFS (PMID: 28067899).

How do the authors explain the potential link of Nos 2 in intracellular with the Vcan (extracellular) ?

Like the Reviewer, we also recognize the importance of elucidating the mediators responsible for connecting the extracellular accumulation of Vcan to the induction of Nos2 expression. In the revised version of the manuscript, we have introduced a hypothesis suggesting that VCAN interaction with cell

surface proteins (including EGFR, CD44 or the β 1 integrin) might activate AKT-mediated signaling pathways leading to NOS2 induction. This hypothesis is now discussed in the new version of the manuscript (page 13, paragraph 2). Identifying and functionally characterizing the specific mediators involved in extracellular signaling by Vcan represents a compelling avenue for future research, and we intend to address this in our forthcoming studies.

Fig 8 there is a problem with the image D on the right panel.

We have not been able to observe any anomaly in the right panel of Fig 8D. We suspect that the Reviewer may not have received the figure in its original high-resolution format. To address this, we have attached below a screenshot of Figure 8D, which was initially submitted, for the Reviewer to view in high quality. Furthermore, in line with the Reviewer previous suggestion, we have also included separate high-resolution images for Elastin, Hoechst and Nos2 or pAKT-S473 stainings in **new Figures EV4 and EV5** (corresponding to Figure 8D and 8F, respectively).

In A, is it possible to quantify the difference of pAkt between non coated and coated with Versican ?

Following the Reviewer suggestion, we have quantified pAkt/Akt levels under both uncoated and V3-coated conditions. These quantification results have been incorporated into the **right panel of Figure 8A**, along with an updated caption in the Figure legend.

In their conclusion, in Fig 9, they drew a schema with ADAMTS1. Are they sure that it is not another ADAMTS responsible of the phenotype ? What are the level of ADAMTS 4 and 5, other versicanases in the aortas of Fbn1 C1041G/+?

We concur with the Reviewer suggestion regarding the potential involvement of other members of the ADAMTS family with aggrecanase activity in MFS aortic disease. Unfortunately, to the best of our knowledge, there are currently no specific and reliable antibodies available for Adamts4 and Adamts5 that would allow for a definitive assessment of their contribution. However, it should be noted that mice lacking the catalytic domain of Adamts5 exhibit increased aortic dilatation upon AngII infusion compared to AngII-treated WT mice (PMID: 29622560), suggesting that Adamts5 could also play a role in MFS aortopathy. We have now addressed these issues in the revised manuscript on page 10, paragraph 3.

In the introduction, line 98, put C1039G/+ as the exponent of Fbn1 and change C1039 to C1041 (the correct number in mouse).

We have corrected this mistake. Thank you.

Referee #2 (Comments on Novelty/Model System for Author):

The mouse model used in this manuscript is widely acceptable and well-characterized that greatly representative of the progression of the aortopathy of this rare disease. In addition, authors use isolated vascular aortic smooth muscle cells to evaluate some mechanistic process.

Referee #2 (Remarks for Author):

The manuscript presented by Redondo and coworkers introduce versican (but not aggrecan) as a new driver of the aortopathy progression in Marfan syndrome taking advantage of respective subproducts generated by ADAMTS1, which is known to be downregulated in Marfan aorta. They nicely utilize new mouse models (Fbn1C1041G/+Vcanhdf/+) and (Fbn1C1041G/+ Acancmd/+) to evaluate their respective impact in the aneurysm formation. Likewise, they specific target to aorta lentiviral tools against the expression of both proteoglycans.

The manuscript is very well written and nice to read. Introduction is clear and rationale of experiments and tools used are also clear.

We thank the Reviewer for his/her positive and constructive comments on the manuscript

I would suggest that in the panel where they represent the experimental design using lentiviral injections (Figs 4B and 8B) they clearly indicate the age at which animals are injected (12 weeks age) regardless it is already mentioned in the figure legend and/or in the text. Many times, readers directly look at the figures to follow results ignoring the reading of figure legends (usually long or very long ones).

We agree with the Reviewer and have now included the age of the mice in **Figures 4B, 6B and 8B**.

Results are very well described and clear. Everything fits very well, but there are only two things that somehow worries me about.

1. The cellular localization of Vcan (fig. 1B) and the nuclear-like staining for the presence of both proteoglycans (Figs 4 and 6), which is not the case in human aortae (EV1). I would expect to visualize the HRP staining at the extracellular space and not intracellularly, and at lesser extent into the nucleus.

We appreciate the Reviewer's concern regarding proteoglycans staining. Although we believe that no firm conclusions can be drawn about the exact localization of staining with immunohistochemistry, we also consider the staining pattern shown is consistent with extracellular staining. We think that the immunofluorescence experiments that we now present to determine at what age the accumulation of Vcan occurs in the aortas of Marfan mice (new Figure EV1) more convincingly reflect the extracellular localization of Vcan in the aorta.

2. The unexpected regression of the aortopathy (both thoracic and abdominal) following lentiviral injections (already evident at the 3rd day after injection and being fully restored after the week/7 days). It would be interesting to have transmission electron microscopy images of aortae before and after lentiviral injections to know in detail to what extent the aortic architecture is normalized. In this respect, ultrasound images do not resolve enough.

We appreciate this Reviewer's concern, and believe there may be a misunderstanding. It seems the reviewer interpreted that Vcan silencing regressed aortopathy in only 3 to 7 days after lentiviral delivery. However, we only show a regression of aortic dilatation in such a short time interval. To clarify the time

elapsed between lentiviral transduction and its effects on the observed phenotypes in aortic wall architecture, we have explicitly indicated this in the text of the revised manuscript in both the Results and Discussion sections (page 8, paragraph 2; page 12, paragraph 2)

In addition to aortic dilation, we have also analyzed other features of aortopathy, including medial thickening and elastic fiber fragmentation and disorganization. It is important to note that recovery of elastic fibers and medial thickness after Vcan knockdown is observed 4 weeks after lentiviral administration, not 7 days (Figures 7 A-C). Furthermore, our previous experience with other mediators of aortopathy in MFS has taught us that aortic architecture does not normalize as early as 7 days after lentiviral administration. For example, *in vivo* inhibition or lentiviral silencing of PKG1 in MFS mice revealed that while aortic dilation also regressed after 1 week of treatment, recovery of aortic architecture took almost 4 weeks (PMID: 33976159).

We agree with the Reviewer that TEM analysis is a valuable approach for assessing aortic architecture. However, we believe that TEM images will yield conclusions similar to those already drawn from Van-Gieson elastic staining.

3. It is rather surprising that the treatment with AZD8055 (inhibitor of Akt signaling) for 2 and 4 days is able to revert the aortopathy, when usually most of pharmacological treatments takes longer to revert aortic injuries in the same MFS mouse model (and in other too mgR/mgR for example). It would be interesting to read a comment in the Discussion in this respect.

As in the case of Vcan silencing, the effect of AZD8055 treatment regresses aortic dilation in 4 days, but the recovery of elastic fibers and medial thickness, as indicated above, takes longer. We are also surprised by how quickly pharmacological inhibitors reverse aortic dilation. We have similarly observed a rapid effect of the AKT inhibitor on the regression of aortic dilation in *Adamts1^{+/-}* mice (PMID: 28067899). Furthermore, we have reported other pharmacological treatments that rapidly regress aortic dilation. For instance, pharmacological inhibition of PKG1 regressed aortic dilation after just 1 week of treatment (PMID: 33976159). Similarly, pharmacological inhibition of sGC completely reversed aortic dilation in MFS mice after only 3 days of treatment (PMID: 33976159). As the Reviewer suggests, we have included commentary on these issues in the revised version (page 13, second paragraph).

Finally, looking the nice EV2 images, it seems rather clear that there is no apparent association between the expression of pAKT-S473 and the level of disarray of elastic laminae. I think that this deserves some commentary in the Discussion.

We agree with the Reviewer that there is no clear association between pAKT-S473 levels and elastic fiber fragmentation and disarray in MFS patients. Following his/her suggestion, we have included a comment in the revised manuscript (page 14, paragraph 1) that discusses how AKT activation in the aortopathy of patients with MFS does not seem to correlate with elastic fiber fragmentation. However, it is possible that such activation is related to aortic enlargement.

Referee #3 (Comments on Novelty/Model System for Author):

A well performed study with different models showing the same outcome, so it feel robust.

Referee #3 (Remarks for Author):

Versican accumulation drives Nos2 induction and aortic disease in Marfan syndrome via Akt activation.

A clear study to find the missing link between the ADAMTS1 and iNOS dysregulation, which the authors published previously. Signs from the literature were already present, yet the dots are now connected by VCAN. I do not have much to add to the data itself, however, a number of interesting references are lacking, where hidden clues on dysregulated VCAN or Akt signaling in Marfan were already provided, which should be mentioned in the discussion.

We thank the Reviewer for carefully reading the manuscript and for providing thoughtful suggestions to improve the revised version. We have included and mentioned all the references he/she suggests in the Discussion (pages are indicated below).

Arterioscler Thromb Vasc Biol. 2023 Jul;43(7):1134-1153. doi: 10.1161/ATVBAHA.122.318448. Page 13.
Arterioscler Thromb Vasc Biol. 2019 Sep;39(9):1859-1873. doi: 10.1161/ATVBAHA.118.312175. Page 11.
Matrix Biol. 2022 Jun;110:106-128. doi: 10.1016/j.matbio.2022.05.002. Page 14.
Clin Epigenetics. 2021 Dec 11;13(1):217. doi: 10.1186/s13148-021-01204-4. Page 14.

How should we interpret the data with regard to the eNOS/Akt study in endothelial cells in MFS mice? Br J Pharmacol. 2007 Apr;150(8):1075-83. doi: 10.1038/sj.bjp.0707181. Could it be the loss of eNOS and Akt signaling there that trigger the SMC to attempt to rescue the lack of NO? In other words, would there be crosstalk as to how the pathological process starts? Is eNOS/pAkt recovered in endothelial cells upon VCAN haploinsufficiency? Can the authors see this in their IHC or IF stainings?

Regarding the onset of aortic disease in MFS, the paper referenced by the Reviewer indicates that eNos (Nos3) activation is impaired in aortas of MFS mice from 6 months (> 24 weeks) of age. Our studies were conducted in 12-week-old mice, and, in agreement with this report, we observed similar phosphorylation levels of Nos3 at Ser1177 in both WT and MFS aortas (**Figure 1 for the Reviewer**). These findings suggest that Nos3 activation is not impaired at this early age. Consequently, the increased aortic expression of Nos2 in MFS mice appears to occur before any impairment of Nos3 activation.

Figure for reviewers removed

Can VCAN (or ACAN) be found in plasma or serum of the mice and is there a correlation between these levels and aortic dilatation? That would strengthen their discussion on this topic.

Following this Reviewer's suggestion, we have measured the aortic diameter of WT and *Fbn1*^{C1041G/+} mice and analyzed Vcan levels in plasma samples from these mice to investigate whether there is a correlation between Vcan plasma levels and aortic dilatation. We measured plasma Vcan levels in mice at 12, 24, and 36-40 weeks of age. We found that this measurement was not reproducible in the plasma of 12 and 24-week-old mice but was consistent in the oldest mice. As indicated in the Discussion of the revised version (pages 11-12), we observed a slight increase in plasma Vcan levels in MFS mice at 36-40 weeks of age compared to age-matched wild-type (WT) mice (**new Appendix Figure S1**). However, we did not find a correlation between plasma Vcan levels and the diameter of the ascending aorta or abdominal aorta (not shown). Nevertheless, it remains possible that in older mice, where the aortic wall deterioration is more advanced, there could be an increase in Vcan presence in the plasma, and plasma Vcan concentration might correlate with the diameter of the ascending aorta. Additionally, in other mouse models of MFS that exhibit more severe pathology or in patients with advanced aortic pathology, a correlation between plasma Vcan levels and aortic diameter cannot be ruled out.

Dear Juan Miguel,

Congratulations on a great revision! Overall the referees are in support of publication. However there remain a few editorial items that we ask you to address in a new revision. In a new point-by-point response please address the following:

1. Please add the following funding information to your online account: RTI2018-548 099246-B-I00, CEX2021-001154-S and CEX2020-001041-S, PI17/00381, PI21/00084, 20151330, INNVAL 21/24, The Marfan Foundation USA555, Faculty grant 2017 MRF/1701, Fundación MERCK-Fundación Española de Enfermedades Raras 2022 and V-Ayudas "Muévete por los que no pueden 2021", and Spanish Ministerio de Ciencia e Innovación contracts FPI (BES-2016-077649); Sara Borrell (CD18/00028) and Juan de la Cierva (IJC2020-044581-I) to M.T.; Ramón y Cajal (RYC2021-033343-I) to J.O.; and FPU (20/04814).
2. Please remove the author contribution section from the main manuscript.
3. Please review our new policy on conflict of interests on the EMBO author guide website and update the title of the section to: Disclosure and competing interests statement.
4. Please provide a summary figure for the synopsis. The size should be 550 wide by 200-440 high (pixels). You can also use something from the figures if that is easier.
5. Please remove the synopsis text from the main manuscript and upload as an individual file.
6. Figure legends: Please note that the box plots need to be defined in terms of minima, maxima, centre, bounds of box and whiskers, and percentile in the legend of figures 2b-d; 3a-c.
7. Our image integrity analysis identified possible re-use of images. (Figure 7D with EV3, Figure 8D with EV4, and 8F with EV5). After reviewing this, I am confident this is an acceptable case of reusing the images, but we do require expressly stating the re-use of figures in the figure legend.

Thank you for the opportunity to consider your work for publication, I look forward to your revision.

Kind regards,
Kelly

Kelly M Anderson, PhD
Scientific Editor
EMBO Molecular Medicine

***** Reviewer's comments *****

Referee #1 (Remarks for Author):

No other comments.
Thank you to the authors for their answers and explanations.

Referee #2 (Comments on Novelty/Model System for Author):

I have also answer to these questions in my previous review

Referee #2 (Remarks for Author):

Authors have satisfactorily responded to my concerns and I see that the other ones as well. It is a good work.

Referee #3 (Remarks for Author):

I have nothing further

The authors addressed the remaining editorial issues.

10th Nov 2023

Dear Juan Miguel,

Congratulations on an excellent manuscript. I am pleased to inform you that your manuscript has been accepted for publication in the EMBO Journal. Thank you for your comprehensive response to referee concerns and for providing detailed source data. It has been a pleasure to work with you to get this to the acceptance stage.

I will begin the final checks on your manuscript before submitting to the publisher next week. Once at the publisher, it will take about 3 weeks for your manuscript to be published online. As a reminder, the entire review process, including referee concerns and your point-by-point response, will be available to readers.

I will be in touch throughout the final editorial process until publication. In the meantime, I hope you find time to celebrate!

Warm wishes,
Kelly

Kelly M Anderson, PhD
Scientific Editor
EMBO Molecular Medicine
